# Sodium enhances indium-gallium interdiffusion in copper indium gallium diselenide photovoltaic absorbers

Diego Colombara [1,7], Florian Werner [1], Torsten Schwarz[2], Ingrid Cañero Infante [3], Yves Fleming[4], Nathalie Valle[4], Conrad Spindler [1], Erica Vacchieri [5], Germain Rey[1], Mael Guennou [4], Muriel Bouttemy[6], Alba Garzón Manjón[2], Inmaculada Peral Alonso [1], Michele Melchiorre[1], Brahime El Adib[4], Baptiste Gault [2], Dierk Raabe [2], Phillip J. Dale [1] & Susanne Siebentritt[1]

Copper indium gallium diselenide-based technology provides the most efficient solar energy conversion among all thin-film photovoltaic devices. This is possible due to engineered gallium depth gradients and alkali extrinsic doping. Sodium is well known to impede inter-diffusion of indium and gallium in polycrystalline $Cu(In,Ga)Se_2$ films, thus influencing the gallium depth distribution. Here, however, sodium is shown to have the opposite effect in monocrystalline gallium-free $CuInSe_2$ grown on GaAs substrates. Gallium in-diffusion from the substrates is enhanced when sodium is incorporated into the film, leading to $Cu(In,Ga)Se_2$ and $Cu(In,Ga)_3Se_5$ phase formation. These results show that sodium does not decrease per se indium and gallium interdiffusion. Instead, it is suggested that sodium promotes indium and gallium intragrain diffusion, while it hinders intergrain diffusion by segregating at grain boundaries. The deeper understanding of dopant-mediated atomic diffusion mechanisms should lead to more effective chemical and electrical passivation strategies, and more efficient solar cells.

[1] University of Luxembourg—Physics and Materials Science Research Unit. 41, rue du Brill, L-4422 Belvaux, Luxembourg. [2] Max-Planck-Institut für Eisenforschung Max-Planck-Straße 1 40237 Düsseldorf, Germany. [3] CNRS, Institut des Nanotechnologies de Lyon CNRS UMR5270 ECL INSA UCBL CPE 7, Avenue Jean Capelle 69621 Villeurbanne, France. [4] Luxembourg Institute of Science and Technology—Materials Research and Technology Department, 41, rue du Brill L-4422 Belvaux, Luxembourg. [5] Ansaldo Energia, Via Nicola Lorenzi, 8 16152 Genova, Italy. [6] Université de Verailles—Institut Lavoisier, 45 avenue des États-Unis 78035 Versailles, France. [7] Present address: International Iberian Nanotechnology Laboratory—Quantum Materials Science and Technology Department, Avenida Mestre Jose Veiga 4715 Braga, Portugal. Correspondence and requests for materials should be addressed to D.C. (email: diego.colombara@bath.edu)

With 22.9% efficiency, Cu(In,Ga)Se$_2$ (CIGS) solar cells show the highest ever reported light-to-electricity conversion of all thin-film photovoltaic technologies[1]. The compound is a solid solution of CuInSe$_2$ (CIS) and CuGaSe$_2$ (CGS)[2], with a band-gap that can be tuned from ca. 1.0 eV (CIS) to ca. 1.7 eV (CGS)[3]. In order to take advantage of a favourable band structure, CIGS devices are typically produced with a v-shaped Ga depth gradient that increases the collection of photogenerated electrons while reducing recombination at the CIGS/CdS interface[4,5]. Recent simulations have shown that the resulting trade-off between short circuit current and open circuit voltage depends on the position of the minimum (notch) in the Ga depth profile[6]. As a consequence, the efficiency could be further improved by shifting the notch position closer to the absorber surface, compared to recent record efficiency devices[7,8].

Until recently[9], the highest reported efficiencies were achieved with a typical Ga content of 0.3. However, having a larger band-gap would bring performance improvement under operation conditions, due to a lower temperature coefficient at maximum power output[10]. Overall, a better control of the Ga diffusion would be most beneficial for CIGS fabrication by selenization of metallic precursor films, also in record cells[11–15], and could pave the way to even higher energy conversion efficiencies.

Besides controlling the Ga depth profile, chemical passivation by extrinsic doping with Na, K, Rb, and Cs seems essential to ensure the best optoelectronic properties in CIGS[9,16–18]. This is normally achieved by diffusion from soda-lime glass or post-deposition treatments (PDTs). However, alkali-metal doping also happens to hinder In/Ga interdiffusion in polycrystalline CIGS films. The effect was observed almost regardless of the alkali metal incorporation source[6,19–26]. This behaviour is ascribed to a reduction of the concentration of Cu vacancies ($V_{Cu}$) following Na addition. Since the diffusion of In/Ga in Cu-poor CIGS is likely to occur via copper vacancies[21,27,28], a decrease of the concentration of copper vacancies is thought to decrease the chances for In and Ga to interdiffuse[28].

From computational studies[29,30], the capture of Na by $V_{Cu}$ and formation of sodium on copper ($Na_{Cu}$) antisite defects in CIGS seems thermodynamically favourable, even considering the most recent $Na_{Cu}$ formation energies calculated with a more reasonable Na electrochemical potential[31,32]. However, it is well known that Na segregates mostly to grain boundaries[33–35]. Therefore, it is difficult to disentangle the effect of Na on the In/Ga interdiffusion inside the grains from that at grain boundaries in polycrystalline films.

Here, the effect of Na PDT on In/Ga interdiffusion is investigated on epitaxial CIS films on GaAs substrates. The advantage is that the system can be treated as a diffusion couple, where diffusion through grain boundaries may be excluded, as it was previously demonstrated by Schroeder et al.[27,36]. The effect of Na on the diffusion of Ga from the substrate through the film is measured by secondary ion mass spectrometry (SIMS), energy-dispersive X-ray spectroscopy (EDS), cross sectional nano-Auger electron spectroscopy (AES) and atom probe tomography (APT). Na incorporation induces a substantial enhancement of In/Ga interdiffusion within the epitaxial CIS films, with respect to a Na-free blank. This is also confirmed by a shrinkage of the unit cell measured by X-ray diffraction. The films remain epitaxial after the treatments, as revealed by electron back scatter diffraction mapping (EBSD), X-ray reciprocal space map (RSM) and scanning transmission electron microscopy (STEM).

The diffusion processes occurring during CIGS growth and PDTs are undoubtedly different[37]. Nevertheless, they are subject to the same laws of diffusion. These results shed light on the mechanism of In/Ga interdiffusion in CIGS and suggest that the grain boundaries play a decisive role by acting as a barrier for intergrain diffusion of In and Ga in conventional polycrystalline CIGS films.

## Results

**Rationale**. In order to investigate the effect of Na doping on In/Ga chemical distribution within Cu-poor epitaxial CIS films on GaAs, three samples are characterised. A CIS film on GaAs subject to annealing with elemental Se and Na$_2$Se vapour (Se+Na$_2$Se) is compared to a CIS/GaAs reference sample exposed to just elemental Se vapour (Se-only), and an untreated CIS/GaAs reference. The comparison with the two references allows to deconvolute the effect of Na incorporation from those due to Se exposure and thermal treatment. Different chemical and structural analyses were performed on these films (SIMS, nano-AES, SEM-EDS, STEM-EDS, EBSD, XRD, APT, spectrophotometry, photoluminescence (PL) and Raman spectroscopy) and the results are discussed here.

**Enhancement of In/Ga interdiffusion**. The SIMS compositional depth profiles are shown in Fig. 1a, b. The $^{23}$Na SIMS depth profiles in Fig. 1a confirm that Na is incorporated into the CIS films via the gas-phase[25], yet the absolute Na concentration for all samples is below the detection limit of EDS (ca. 1 at.%). This is confirmed by a maximum concentration of ca. 0.4 at.% measured by APT for sample Se+Na$_2$Se. An increase of the overall $^{23}$Na SIMS signal is observed (more than 250-fold) upon sodium PDT (Se+Na$_2$Se), compared to untreated and Se-only films. A small Na incorporation at the surface and at the back is also observed for Se-only, i.e., in the absence of intentional Na sources, and is attributed to trace Na impurities in the Se employed[25]. The rapid Na incorporation in epitaxial CIS is consistent with the low activation energy of Na diffusion reported recently for single crystal CIS[38].

The Ga compositional depth profile is estimated as the $^{69}$Ga/($^{115}$In + $^{69}$Ga) SIMS (GGI) signal ratio (Fig. 1b). At the back interface of the CIS film the ratio increases by about one order of magnitude after the PDT with just selenium, compared to the untreated film. This implies that Ga in-diffusion from the GaAs substrate occurs appreciably under such conditions at 570 °C. However, the Ga content at the surface of the film is nearly unaffected (below 0.01), thus the film is strongly graded. The Na PDT increases the In–Ga interdiffusion substantially compared to the untreated sample. In this case, the GGI increases by about two orders of magnitude at the back interface and more than one order of magnitude at the front. As a result, this film is also strongly graded, but the overall Ga in-diffusion is much more pronounced. The effect is reinforced by the fact that the Se+Na$_2$Se film is actually slightly thicker than the untreated and Se-only films (1.2 vs. 0.8 μm).

SEM-EDS analysis was performed at 7 kV, which results in >90% of the In/Ga L X-ray signal arising from ca. the topmost 200 nm of the films (i.e., without Ga contribution from the GaAs substrate). The corresponding GGI ratios are: 0%, 0.8% and 7% for untreated, Se-only and Se+Na$_2$Se samples, respectively, (data included as black circles in Fig. 1c). EDS analyses up to 13 kV provide a consistent picture and are available in Supplementary Fig. 1, along with Casino Monte Carlo simulations[39].

The in-diffusion of Ga into the CIS film is counterbalanced by out-diffusion of In into the GaAs, i.e., In/Ga interdiffusion takes place at the interface of the diffusion couple. The In SIMS signal in the GaAs substrate is proportional to the Ga signal in the CIS films, as shown in Fig. 1c. The proportion is not 1:1 because SIMS is not an absolute quantitative technique in the absence of standard samples[40]. Furthermore, Ga in-diffusion in CIS is faster than In out-diffusion in GaAs, consistent with Kirkendall void accumulation, as seen in the STEM analysis of the cross sections[27].

Nano-AES analysis of the cross sections provides a more quantitative (albeit surface sensitive) estimation of the Ga in-diffusion in CIS. The AES results are a convolution of chemical

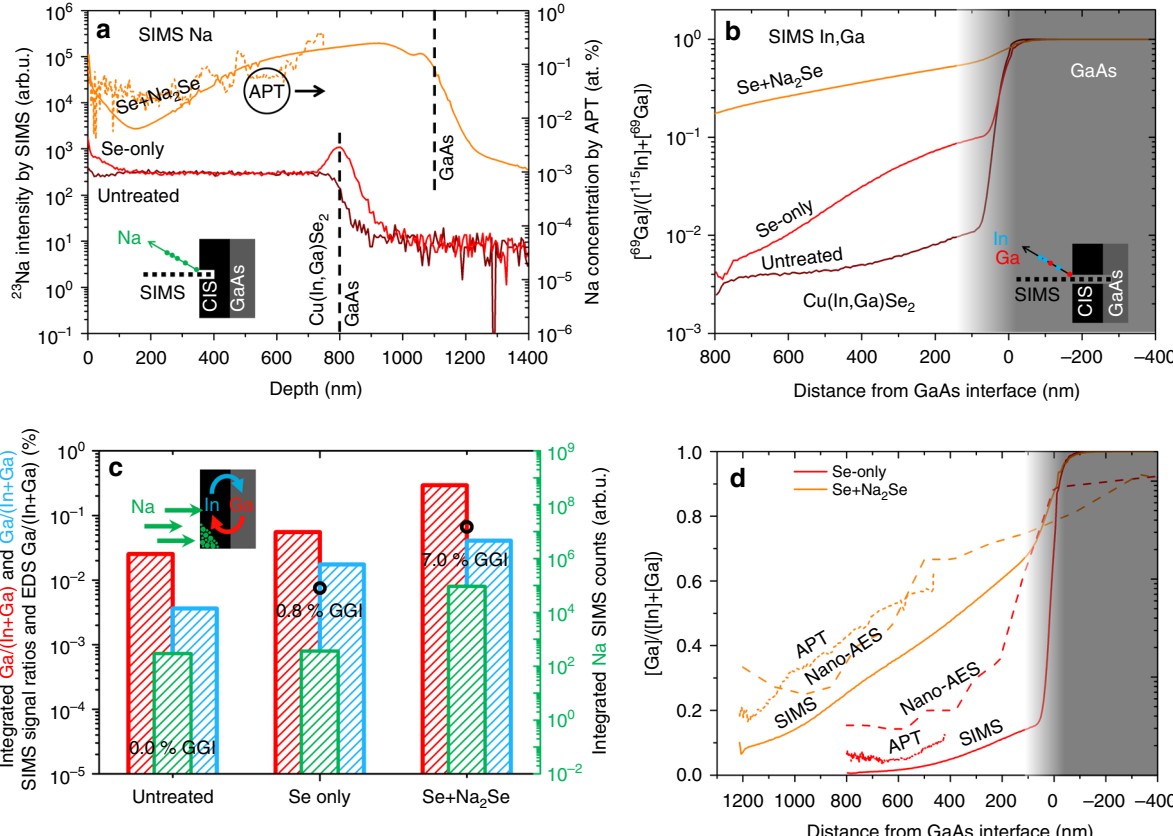

**Fig. 1** Compositional depth profiles of the CIS/GaAs samples subject to different treatments. **a** $^{23}$Na SIMS profiles (left axis) and absolute Na concentration from APT (right axis, only for Se+Na$_2$Se sample) expressed as a function of depth from the CIS surface (note the different film thickness); **b** corresponding Ga depth gradient expressed as $^{69}$Ga/($^{115}$In + $^{69}$Ga) SIMS signal ratio as a function of distance from the CIS/GaAs interface. **c** Extent of Na in-diffusion in the CIS films from the surface until the GaAs interface expressed as integrated $^{23}$Na SIMS counts (green bars). Corresponding integrated $^{69}$Ga/($^{115}$In + $^{69}$Ga) SIMS signal ratios divided by the film thickness (red bars). Extent of In out-diffusion from the films into the GaAs substrate expressed as $^{115}$In/($^{115}$In + $^{69}$Ga) integrated SIMS signal ratio until 500 nm deep in the substrate from the interface (blue bars). The black circles and numbers are the GGI ratios measured by SEM-EDS with an acceleration voltage of 7 kV (corresponding to more than 90% of In/Ga L X-rays arising from first 200 nm depth according to Casino Monte Carlo simulations[39]). **d** GGI depth profiles estimated by correcting the SIMS profiles in **b** by the natural isotopic abundances of In (95.71%) and Ga (60.11%) (solid lines), by nano-AES cross sectional analysis (dashed lines) and APT (dotted lines, the depth range is limited by the sample preparation)

composition and morphological properties of the cross sections' uneven surfaces that have been deliberately analysed as-cleaved. The comparison between AES and SIMS GGI depth profiles shown in Fig. 1d reveals that SIMS underestimates the absolute GGI ratio in CIS, but the trends are consistent. Additional APT analysis of the Se+Na$_2$Se sample confirms at the nanoscale the GGI gradient measured by SIMS and the higher GGI content estimated by nano-AES.

**Microstructural analysis.** The SEM and STEM microstructural analysis is shown in Figs. 2 and 3. The top view SEM images reveal a roughly unchanged surface morphology after the Se-only PDT (Fig. 2a). Conversely, the Se+Na$_2$Se sample clearly shows some cracks (Fig. 3a) attributed to compressive strain relief resulting from the replacement of In by smaller Ga, as well as from the Kirkendall void formation revealed by STEM cross section in Fig. 3e, f (triangular-shaped voids), which is much more pronounced compared to the Se-only film. As a result, untreated and Se-only films are much denser than Se+Na$_2$Se (see Supplementary Figs. 2 to 4). The cross sectional EDS maps visibly confirm that the Se+Na$_2$Se film displays a higher In/Ga inter-diffusion than Se-only. The region enclosed by the white circle seen from different tilt angles reveals the presence of dislocations

in the untreated film, as the contrast of the defects changes with the tilt. More detailed analysis and comparison of such defects in the other samples is given in section APT STEM and SIMS defect analysis and in Supplementary Figs. 5 to 7.

The morphology and texture of the back side after stripping the films from the substrate are shown in Figs. 2 and 3b, c, d as secondary (SE) and backscattered (BSE) electron images and EBSD inverted pole figure maps (IPF) overlaid to the corresponding image quality (IQ) maps. The lift-off procedure induces severe cracking of the films, but allows confirming that the interface remains epitaxial after the In/Ga interdiffusion, as texture deviations are confined to cracks and voids. The SE and BSE images of the back side of Se-only reveal a flat morphology, while small craters are visible in Se+Na$_2$Se, consistent with the aforementioned voids.

**Crystallographic analysis.** Generally, epitaxial films grow either by fully straining the in-plane parameter, in order to minimise the lattice mismatch, or by completely or partially relaxing the misfit strain, thus approaching the crystal structure of the bulk. The in-plane strains are associated with compensating opposite strains of the out-of-plane parameters, assuming conservation of the unit cell volume.

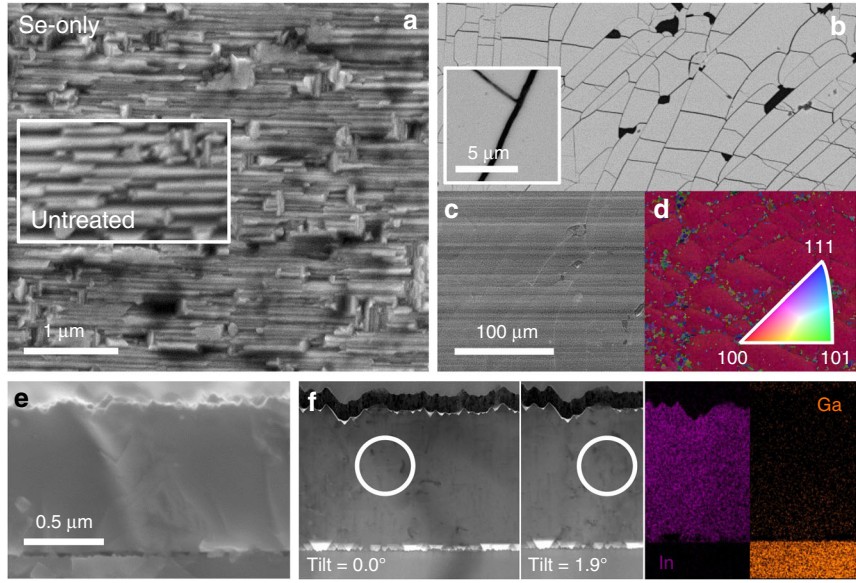

**Fig. 2** Microstructural analysis of Se-only film. **a** SEM SE top view images. **b** BSE, **c** SE images and **d** IPF + IQ EBSD maps of the back side of the films on resin after lift-off. **e** Cross sectional images acquired by SEM and **f** bright-field STEM with corresponding EDS In and Ga mapping

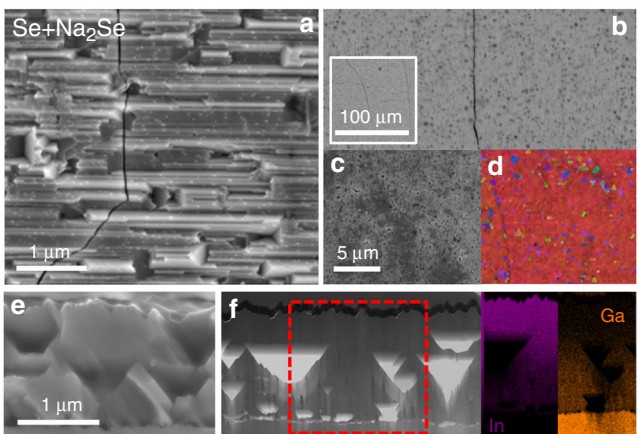

**Fig. 3** Microstructural analysis of Se+Na$_2$Se film. **a** SEM SE top view images. **b** BSE, **c** SE images and **d** IPF + IQ EBSD maps of the back side of the films on resin after lift-off. **e** Cross sectional images acquired by SEM and **f** bright-field STEM with corresponding EDS In and Ga mapping

Bulk CIS and CGS crystallise in the tetragonal space group (I-42d, No. 122), whereas GaAs substrate is cubic (F-43m space group, No. 216). The unit cell parameters of bulk CIS and CGS are $a_{CIS} = 5.784$ Å, $c_{CIS} = 11.621$ Å and $a_{CGS} = 5.614$ Å, $c_{CGS} = 11.022$ Å, and the unit cell parameter of GaAs is $a_{GaAs} = 5.654$ Å. Therefore, from the lattice mismatch of these compounds, the epitaxial growth of CIS on GaAs leads to a compressive in-plane misfit strain of -2% (see Supplementary discussion 1 and Supplementary Figs. 8 and 9)

Bragg-Brentano, RSM and EBSD analyses were performed in order to assess crystallographic orientation, strain and structural modifications induced by Na incorporation. Henceforth, the Miller notations of the reflections are assumed for cubic GaAs and tetragonal CIS unit cells. Fig. 4d shows the Bragg-Brentano diffractograms of samples and GaAs substrate (black curve). The {200} reflection of the GaAs is clearly seen for all samples and has a lower intensity for Se+Na$_2$Se, consistent with its higher thickness compared to untreated and Se-only. Since the unit

cells of CIS and CGS are almost double cubes with lattice parameter similar to GaAs, reflections from {200} of GaAs, CIS and CGS, as well as {004} of CIS and CGS occur within 3° of 2θ. The calculated positions of bulk CIS and CGS reflections are shown as dashed lines, along with the expected reflections of fully strained films (dotted lines). The reflection of the untreated film at ca. 30.7° is consistent with that of bulk {200} CIS. Therefore, it is assumed that CIS grows with the *a*-parameter perpendicular to the substrate (*a*-orientation) and shows negligible strain (cf. lower schematics in Fig. 4b). Clear doublets are also revealed, corresponding to Cu Kα$_1$ and Kα$_2$, which is consistent with a fully epitaxial film, albeit relaxed.

The evolution of the out-of-plane unit cell parameter *a* after the PDTs is due to a convolution of In/Ga interdiffusion and Na in-diffusion. The reflection of Se-only film occurs at ca. 31.3°, i.e., at higher 2θ angular values compared to the untreated film. Also Se+Na$_2$Se shows a reflection at larger angular values, but an additional reflection appears at ca. 32.0°, close to the theoretical value of bulk {200} CGS. A separate XRD analysis from the back side of Se+Na$_2$Se has been performed after lift-off (orange dotted line) and reveals a higher intensity of the additional reflection at ca. 32.0°.

The positive shifts of the {200} reflections after the PDTs imply a decrease of the out-of-plane unit cell parameter. This signature may originate from an increase of the GGI ratio and/or the relief of epitaxial strain, both factors operating through the film thickness. Both Se-only and Se+Na$_2$Se samples lose the Cu Kα$_{1,2}$ doublet due to broadening of the lattice constant distribution. Overall, the angular shifts to higher 2θ and the broadening of the lattice constant distribution are consistent with the GGI depth profiles in Fig. 1.

The GGI ratio in the CIGS films can be determined from the 2θ position of the {200} reflection maxima, assuming a linear variation of the cell constants (Vegard's law) for the Cu(In,Ga)Se$_2$ solid solution[41]. This determination yields a GGI ratio of 0.03 for Se-only and 0.31 for Se+Na$_2$Se. Additionally, the diffractograms of Se+Na$_2$Se are consistent with the presence of a CGS phase (GGI equals 1) towards the back of the film.

In order to shed light on the unit cell deformations and disentangle the contribution of strain, Fig. 4c shows the reciprocal space maps (RSMs) including the {10 1 1} crystallographic planes

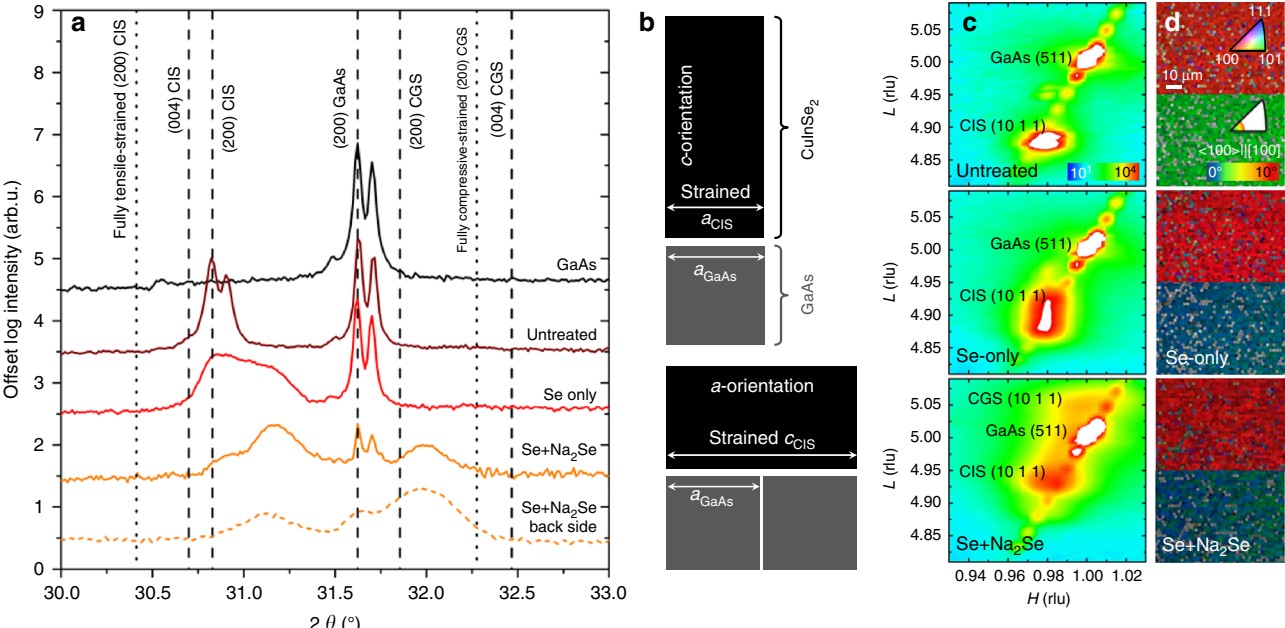

**Fig. 4** Crystallographic analysis of the CIS/GaAs samples subject to different treatments. **a** High-resolution XRD analysis in $\theta$–$2\theta$ configuration. The reference reflections in dashed lines are calculated from ICDD of GaAs 00-032-0389[62], Cu-poor CuInSe$_2$ 01-070-3356[63] and CuGaSe$_2$ 01-076-1735[64] (for clarity, only K$\alpha_1$ = 1.54056 Å reflections are shown). The dotted lines correspond to the estimated positions of the {200} planes of CIS and CGS if the epitaxial films were fully strained, assuming unit cell volume conservation, as described by the schematics shown in **b**. The RSMs of the films are shown in **c**, with H and L indices of the reciprocal lattice units of GaAs {511} and CI(G)S {10 1 1}. For the sake of clarity, the intensity scale has been chosen to enhance the film's reflections and thus the reflection of GaAs appears saturated. **d** EBSD analysis acquired at high pattern resolution with IPF (upper) and corresponding maps showing the deviation from the <100> crystallographic direction (lower)

of the three films, along with the {5 1 1} reflection of the GaAs substrate. The RSM analysis is performed on these specific planes to ensure an appropriate balance of diffraction intensity, deconvolution of in-plane and out-of-plane directions and reciprocal space separation of substrate and film reflections. The relative positions of the reflections of film and substrate allow determining the in-plane and out-of-plane parameters through the thickness of the films. Remarkably, the RSMs reveal unique spots for K$\alpha_1$ and K$\alpha_2$ of the {5 1 1} GaAs substrate (the reciprocal space is referred to the H and L positions of the {5 1 1} K$\alpha_1$ reflection of GaAs). The low-intensity diagonal pattern present in the three maps is a measurement artefact due to detector streak interference arising from air scattering.

The film reflections inspected in the RSMs correspond to {10 1 1} planes of CI(G)S. Given the crystallographic orientation of the films, the H Miller index corresponds to the out-of-plane direction and L to the in-plane direction. The in-plane projection of the reflections indicates a unique in-plane component, suggesting that the treated films retain their epitaxial nature, or possess a very small degree of mosaicity. Moreover, the H value (H equals 0.98) is the same for all samples, indicating a slightly relaxed in-plane parameter. The PDTs affect the out-of-plane projection of the {10 1 1} reflection, with a vertical projection broadening in L. This result is consistent with the broadening observed in the {200} reflections in the $\theta$–$2\theta$ scan (Fig. 4a), with a L scaling broadening towards higher values attributed to In/Ga interdiffusion. For Se+Na$_2$Se, the {10 1 1} reflection moves to higher L values and a second smaller spot appears at even higher L values than the corresponding one for GaAs, which is consistent with the reflection in $\theta$–$2\theta$ scan attributed to bulk CGS at the back of the film. Importantly, these RSMs confirm that there is no change of the in-plane unit cell parameter of the films after the PDTs, thus the films remain epitaxial. This happens despite a clear phase segregation in sample Se+Na$_2$Se.

The top surface EBSD maps acquired at high pattern resolution (Fig. 4d) reveal a very low deviation from the <100> crystallographic direction in all films, regardless of the treatment. The deviations averaged over ca. 5000 μm$^2$ are: 2.1°, 1.0° and 0.9° for untreated, Se-only and Se+Na$_2$Se, respectively.

**Raman and optical analysis**. Raman and optical analysis have been performed to provide a complementary chemical identification of the phases present in the films, especially at the back interface. Fig. 5 shows the Raman (a) and photoluminescence (b) spectra of Se-only and Se+Na$_2$Se films at the interface with GaAs after lift-off from the substrate. Se-only shows the main A$_1$ peak of CIGS and a second smaller peak at lower frequencies. The first peak could be erroneously attributed to a CIGS with GGI ca. 0.4, based on its position[42]. However, a dedicated EDS analysis at 7 keV reveals ca. 0.15 GGI ratio and ca. 0.90 Cu/(In + Ga) (CGI) ratio, meaning that the peak occurs at higher frequencies than expected for Cu stoichiometric compositions, due to Cu deficiency[43,44]. The second peak is attributed to the In-pure ordered defect compound (ODC) phase with 1:3:5 stoichiometry CuIn$_3$Se$_5$[45]. Se+Na$_2$Se displays mostly a broad peak typical of an ODC Cu(In,Ga)$_3$Se$_5$ with GGI ca. 0.7, whose presence is confirmed by the ca. 0.67 GGI and ca. 0.46 CGI ratios measured by EDS at 7 keV on the back side.

The PL spectra in Fig. 5b reveal a single broad peak centred at 1.03 eV for Se-only and a convolution of 1.03 and 1.23 eV peaks for Se+Na$_2$Se. Optical absorption measurements through the whole CIGS/GaAs stacks yield absorption onsets at 1.02 and 1.18 eV for Se-only and 1.20 eV with a shoulder at lower photon energies for Se+Na$_2$Se. The low band-gap onset and PL peaks are consistent with the CIS phase, while the wider band-gap onsets and PL peak are consistent with the ODC phase with variable GGI ratio[46].

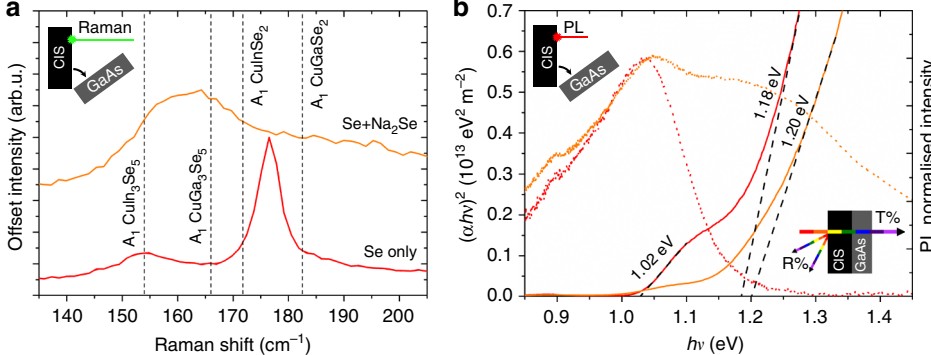

**Fig. 5** Raman and optical characterisation of the Se-only and Se+Na$_2$Se samples. **a** Raman spectra (532 nm) of the back sides of Se-only and Se+Na$_2$Se films after lift-off and **b** corresponding photoluminescence spectra (dotted curves, room temperature, 16 mW laser power). The GGI dependence of the A$_1$ Raman frequencies of CuInSe$_2$, CuGaSe$_2$[42], CuIn$_3$Se$_5$ and CuGa$_3$Se$_5$[45] are shown as dashed vertical lines in **a**. The solid curves in **b** are the optical absorption spectra obtained by illumination from the front surfaces through the CIS/GaAs samples

The Raman, PL and EDS analyses of the back interfaces of Se-only and Se+Na$_2$Se films are consistent with the SIMS, EDS and nano-AES analyses shown in Fig. 1, supporting the observation that Na incorporation enhances Ga in-diffusion from the GaAs substrate. No evidence of CGS formation is revealed by PL and Raman for the Se+Na$_2$Se film. Instead, it appears that Ga in-diffusion occurs to the extent that a Ga-rich Cu(In,Ga)$_3$Se$_5$ ODC phase forms at the back interface. The weak reflection in the XRD at 32.0° attributed to CGS in Fig. 4a may actually arise from the {200} planes of the Cu(In,Ga)$_3$Se$_5$ ODC phase[47].

**APT STEM and SIMS defect analysis.** Fig. 6 shows complementary three-dimensional APT and SIMS analyses of the three films with a focus on Cu and Na distributions within Se+Na$_2$Se. Two specific regions of interest in the untreated and Se+Na$_2$Se APT maps are identified by red circles and discussed separately (Fig. 7). Three-dimensional videos of the APT maps are also available (see Supplementary Movie 1).

The two orthogonal views of the untreated sample reveal clearly the presence of a planar defect. This is most likely a stacking fault, because it is constrained within the APT data set. Notably, both untreated and Se-only films contain regions with Cu concentration higher or lower than 19 at.%, as shown by the Cu isosurfaces.

The overall Cu concentration of the Se+Na$_2$Se APT specimen is lower. Here, Cu isosurfaces are drawn at 15 at.% (blue) and overlaid with the elemental distribution of Na (green). Figure 6b shows the corresponding $^{63}$Cu (solid blue) and $^{23}$Na (solid green) SIMS signal depth profiles of Se+Na$_2$Se averaged over a larger sample area (ca. 100 μm x 100 μm). The Cu compositional profile is highly disrupted, with a relative decrease of 30% from near the front to the back of the film. It is excluded that the depth distribution of Kirkendall voids in the film has a strong influence on the SIMS profiles because the matrix variations are taken into account by Cs$^+$ normalisation (cf. Methods section). By comparison, the $^{63}$Cu SIMS profiles of untreated (dotted blue) and Se-only (dashed blue) films are more uniform.

Figure 6d provides the calculated composition of regions I–VI within the Se+Na$_2$Se APT specimen. A nearly monotonic decrease of In and increase of Ga is observed from region I to VI, which is consistent with Figs. 1 and 3. Two regions with lower Cu content are identified: at the surface (region I) and deeper in the film (region V), consistent with SIMS. The APT data strongly indicates a higher Na concentration in these Cu-poor regions. This is not surprising given that Cu and Na can be isoelectronic, i.e., Na atoms can replace Cu and form Na$_{Cu}$ antisites[28]. However, APT reveals that the Na:Cu replacement is much lower than 1:1.

The concentration of Na would have to be approximately 22 times higher in order to compensate for the lack of Cu in these Cu-poor regions and attain the (Cu,Na)(In,Ga)Se$_2$ 1:1:2 stoichiometry. Indeed, these two regions have higher In and Se concentration than the neighbouring material portions. Their composition is compatible with the ODC phase with variable In/Ga substitution. Region V approaches the composition Cu(In,Ga)$_3$Se$_5$ with In:Ga equals 1:1, consistent with the Raman analysis in Fig. 5a. These ODC regions have a slight deficiency of Se with respect to the 1:3:5 stoichiometry, which may be due to a preferential loss of Se during the APT measurement, a known issue in the analysis of compound semiconductors[48,49].

Note that Na and Cu have very different sensitivity factors in SIMS, so quantitative compositional considerations are not possible with SIMS. Nevertheless, the two SIMS profiles are consistent with the quantitative APT data, plotted on the same scale for comparison. Note that the region of film closer to the GaAs substrate could not be analysed by APT, due to fracture of the specimen closer to this interface. Figure 6c shows the depth profile of [Na]/([Cu] + [Na]) SIMS signal ratio (black) and APT atomic ratio (purple). The profiles confirm quantitatively the increase of Na concentration towards the back of the film, notwithstanding the relatively large fluctuations in the sample that are revealed by APT thanks to its much finer spatial resolution and are instead averaged by SIMS.

Assuming sufficient Na availability during the PDT and no losses of Na-bearing species to the gas-phase during APT sample preparation[25], the Na concentration observed in the different regions could correspond to the respective Na solubility limits. It is striking to observe that the Na concentration in the Ga-rich ODC region V is 20 times higher than in the Ga-poor ODC region I. This result is noteworthy, given that region V is actually deeper in the film compared to region I, i.e., it implies a longer diffusion path for Na. According to the SIMS measurements, the same is true for the Na concentration in the 1:1:2 phase near the surface compared to the Ga-rich CIGS at the back of the film (Fig. 6c). Both APT and SIMS observations suggest an affinity between Na and Ga in CIGS.

A recent APT study of polycrystalline CIGS films suggests that higher GGI ratios may lead to higher Na concentration in the films, due to a higher density of grain boundaries per unit volume[50]. A correlation between Na and Ga concentrations was also shown for polycrystalline CIGS films obtained by selenization of Na-containing metallic precursors[51]. Here, further evidence for the spatial affinity between Na and Ga is gained by plotting the $^{23}$Na/($^{63}$Cu + $^{23}$Na) against the $^{69}$Ga/($^{115}$In + $^{69}$Ga) SIMS depth profile data of the epitaxial films, as shown in Fig. 7a.

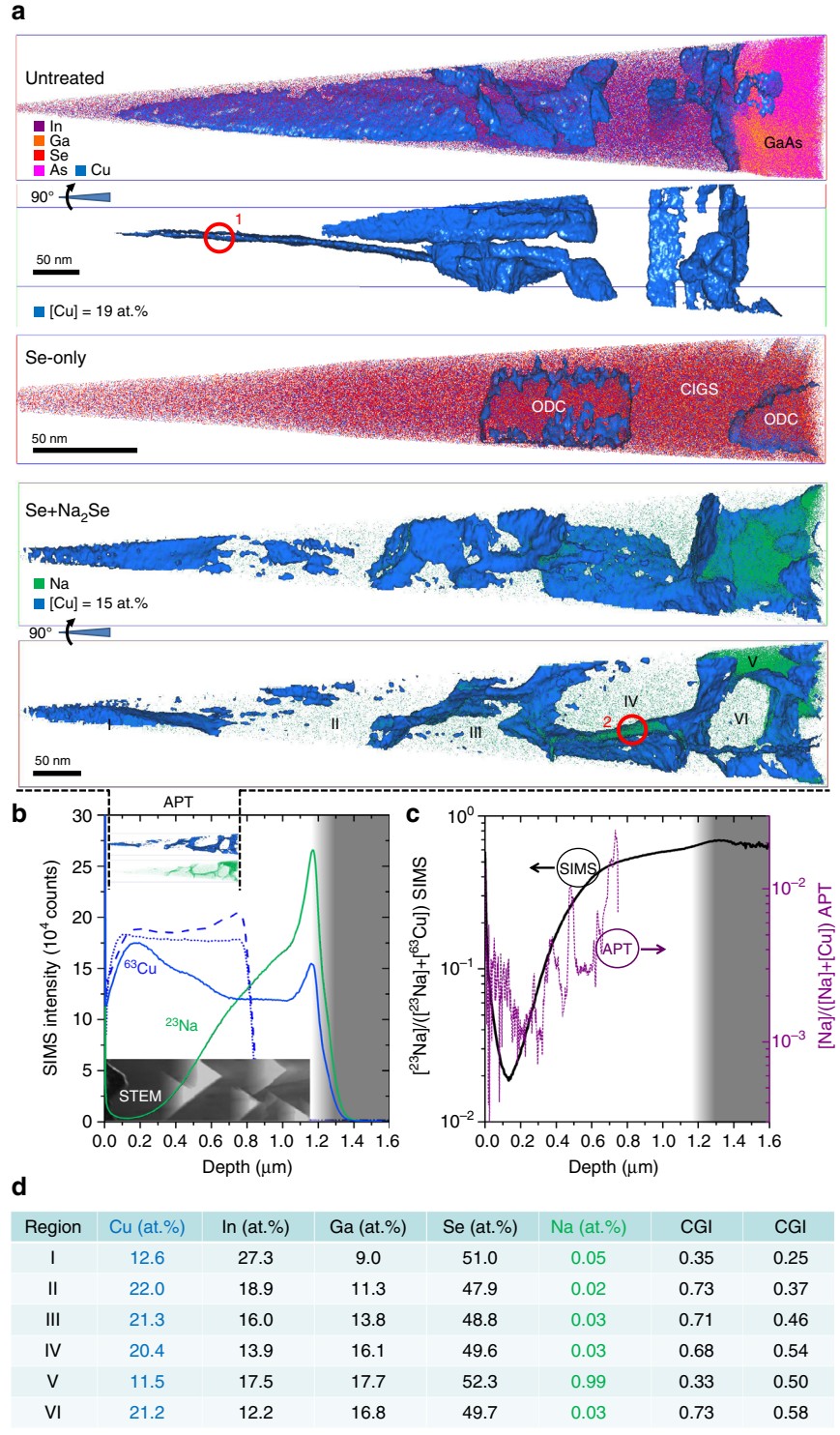

**Fig. 6** APT analysis and SIMS comparison. **a** APT analysis of the three films as seen from different orthogonal views. Cu isosurfaces (blue) at 19 at.% ([Cu] = 19 at.%) for untreated and Se-only and at 15 at.% ([Cu]= 15 at.%) overlayed with Na distribution (green) for Se+Na$_2$Se. The composition of the regions labelled with roman numerals is shown in the table. The two regions enclosed by the red circles are discussed in the main text. **b** Scaled $^{63}$Cu (blue) and $^{23}$Na (green, Se+Na$_2$Se) SIMS signal depth profiles and APT maps and STEM cross section of Se+Na$_2$Se plotted on the same scale and position for comparison. **c** [$^{23}$Na]/([$^{63}$Cu] + [$^{23}$Na]) SIMS signal depth profile ratio (black) and corresponding [Na]/([Cu] + [Na]) atomic depth profile ratio obtained from APT (purple) of Se+Na$_2$Se. **d** Averaged APT elemental quantification of different regions of the Se+Na$_2$Se tip identified in **a** from the surface (I) towards the back of the film (VI)

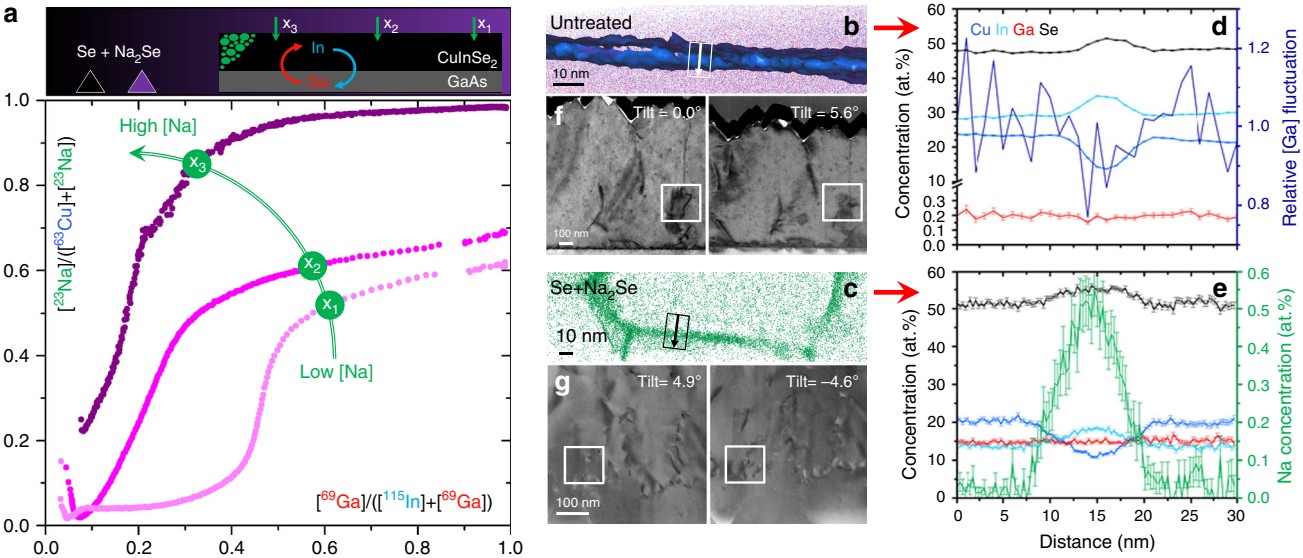

**Fig. 7** Na-Ga affinity and defect analysis. **a** $[^{23}Na]/([^{63}Cu] + [^{23}Na])$ SIMS signal ratio plotted as a function of the corresponding $[^{69}Ga]/([^{115}In] + [^{69}Ga])$ for three locations in the Se+Na$_2$Se sample with increasing Na concentrations (curved green arrow) resulting from a decreasing distance from the sodium selenide source during the PDT (numbered green arrows in schematics). Magnified views of the regions of interest in the APT maps of untreated (**b**) and Se+Na$_2$Se films (**c**) and **d–e** corresponding concentration profiles along the squared arrows. **f–g** STEM images of typical detects encountered in the films taken at different tilt angles

Fig. 7a shows that film regions with high [Ga]/([In] + [Ga]) ratio generally exhibit high [Na]/([Cu] + [Na]) ratio. Regions of the film originally located closer to the Na source during the PDT (i.e., with higher total Na concentration) attain higher Ga incorporation but display a less pronounced spatial correlation between Na and Ga contents, as highlighted by the curved arrow in Fig. 7a. Na diffuses from the surface of the films and tends to accumulate in the Ga-richer areas towards the back of the film. Together with the results shown in Figs. 1, 2, 3, 4, 5 and 6, this is an indication that Na affects the Ga in-diffusion process. The higher affinity displayed by Na and Ga at lower Na concentrations suggests the existence of an enthalpic driving force for the formation of Na+Ga-containing defect clusters. Magnified APT views of the regions of interest enclosed by the red circles in Fig. 6a are now assessed with the intent to better understand the role of Na on the enhancement of In/Ga interdiffusion in CIGS (Fig. 7b, c). The concentration profiles in these regions are compared with representative STEM images (Fig. 7f, g) to help unravelling the nature of such defects.

Region 1 is located around what appears to be a stacking fault in the untreated film. The concentration profile reveals a Cu depletion and In-Se enrichment at the defect spanning ca. 10 nm. Ga has a low concentration and is homogeneously distributed across the defect's interfaces. The Na concentration across the interface is at the background level of ca. 20 ppm. Representative STEM images acquired at different tilt angles confirm that the untreated film is not free from defects, but several linear defects such as dislocations are present. Such defects are also present in the the Se-only film (cf. Fig. 3f). It is common for single crystals to contain subgrains with very low misorientation; stacking faults and twinning have been reported for epitaxial CIS films[52].

Region 2 is located around what appears to be a planar defect approximately at the centre of the Se+Na$_2$Se film. The concentration profile is similar to the untreated case (except the overall higher Ga concentration), but here a pronounced Na enrichment is clearly observed. The defect is surrounded by regions with typical 1:1:2 composition, like for the untreated film. The STEM images show a dense network of defects such as dislocations and/or low-angle grain boundaries. However, the

presence of high-angle grain boundaries typical of polycrystalline films is excluded. Concentration profiles across the interfaces between other regions within the same APT data set confirm these compositional fluctuations but also reveal a slight Ga depletion (Supplementary Fig. 10). Furthermore, the Na interfacial excess varies by 0.19–1.82 atoms/nm$^2$ from one interface to another, with no obvious correlation with the depth of the interface. Overall, the analysed concentration profiles show the presence of Na-rich regions, which are likely to contain a higher density of Na-related point defects. Similar defects with higher Ga concentrations may be present in the Ga-richer part of the film. Confirmation in these regions of the film could not be obtained due to failure of APT specimens close to the CIGS/GaAs interface. Similar compositional variations at planar and linear defects have been observed in CIGS in other APT studies[53], but it is still unclear what favours Ga over In enrichment at such defects.

## Discussion

Since the outset of extrinsic alkali doping in CIGS, Na is known to hinder In/Ga interdiffusion when present during even the last stages of film growth. This work shows that the phenomenon takes place only in polycrystalline films. Here, Na introduced after growth is shown to enhance In/Ga interdiffusion in CIGS films free from grain boundaries. The process is so enhanced as to induce formation of ODC, the contiguous compound in the group III-rich side of the phase diagram, at the interface with GaAs.

Fig. 8 provides a mechanistic description of the phenomenon based on all data collected and drawn directly on STEM-EDS elemental maps. As a consequence of the asymmetric diffusivity of In and Ga, Kirkendall voids form at the interface with GaAs and accumulate into bigger voids to minimise surface energy. Ga in-diffuses into the formerly pure CIS and converts it into CI(G)S. When the concentration of group III elements exceeds the saturation limit, the ODC phase starts to grow out of the CI(G)S. Such a growth may be pseudotopotactic, because the interface is not coherent, but no grain boundaries are detected between the two phases and there is no discontinuity of Ga composition.

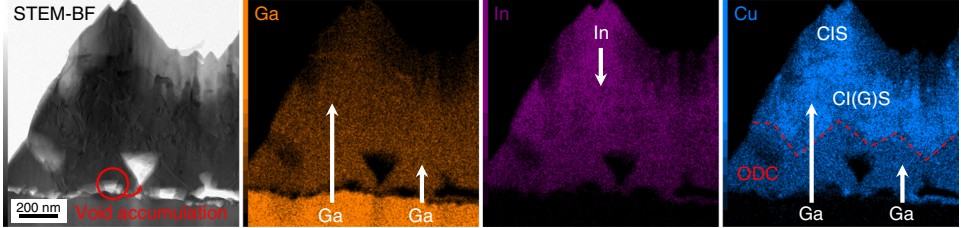

**Fig. 8** Mechanistic description of In/Ga interdiffusion. Bright-field STEM image of a cross section of Se+Na$_2$Se sample and corresponding Ga, In and Cu EDS elemental maps

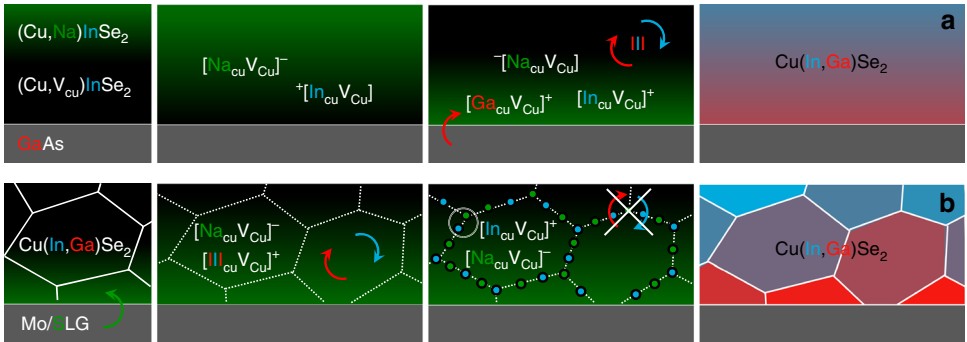

**Fig. 9** Effect of Na on In/Ga interdiffusion. Schematic representation of the effect of Na on **a** enhanced intragrain In/Ga interdiffusion, as seen here in CIS/ GaAs films free from (high-angle) grain boundaries, and **b** hindered intergrain In/Ga interdiffusion, commonly observed in polycrystalline CIGS films. Na: green, In: blue, Ga: red. This hypothetical model considers accumulation of Na and Na-In-Cu defects at the grain boundaries[65]

The dense network of dislocations and/or low-angle grain boundaries in the Se+Na$_2$Se film may have formed as a result of Na solubility saturation (cf. Fig. 6a). Fig. 7 reveals that some of such defects pre-existed even in the untreated film and it is likely that they incurred Na decoration/extension, during the PDT. The role of dislocations on the enhanced diffusion of In and Ga cannot be excluded here, but a recent study on single crystal CIS suggests that dislocations do not control the Na diffusion mechanism[38]. Presumably the enhancing effect of Na on In/Ga interdiffusion occurs also at the interior of CIGS grains in conventional polycrystalline films. Point defect pairs of the type $[III_{Cu}V_{Cu}]^+$ have been historically invoked in order to account for the large tolerance of Cu-poor CIGS to off-stoichiometry, which ultimately leads to ODC phase formation[54,55]. The computational work of Pohl and Albe suggests a more modest (yet favourable) enthalpic stabilisation associated to this defect pair[30], which is therefore predicted to exist in CIGS at equilibrium, based on statistical thermodynamics.

The energetics of Na-containing defect complexes has been computed by Oikkonen et al.[29]. Defect pairs of the type $[Na_{Cu}V_{Cu}]^-$ and $[Na_{Cu}2V_{Cu}]^{2-}$ were estimated to have a null or slightly negative binding energy, but were suggested to be dynamically stable.

Here, defect supercomplexes of the type $\{[III_{Cu}V_{Cu}]^+[Na_{Cu}V_{Cu}]^-\}$ are proposed to form due to the additional coulombic interaction arising from the opposite charge of its component defect pairs. The defect identified by APT in Fig. 7b, c is an indication that such a super complex could exist for In: $\{[In_{Cu}V_{Cu}]^+[Na_{Cu}V_{Cu}]^-\}$. Given that $[Ga_{Cu}V_{Cu}]^+$ is predicted to be even more stable than $[In_{Cu}V_{Cu}]^+$[30], the corresponding Ga-containing super complex $\{[Ga_{Cu}V_{Cu}]^+[Na_{Cu}V_{Cu}]^-\}$ seems also likely, but proving its existence experimentally is far from trivial.

The formation of such Na-containing defect clusters may provide a driving force for enhanced In/Ga interdiffusion in CIGS free from grain boundaries, so presumably also in the interior of the CIGS grains in polycrystalline films. The defect cluster may act as a mediator for dynamic exchange of In and Ga atoms in the Cu sublattice. This possibility is intriguing, given that Stanbery et al.[56] previously suggested that Na destabilises $[In_{Cu}2V_{Cu}]^0$ defect complexes and rejects excess In when used as a dopant in CIS. The APT analyses show Na and In but no Ga enrichment at the planar defects observed in the film. Grain boundaries of polycrystalline films tend to show In[50] or Ga[57] enrichment, along with substantial amounts of Na.

Two related hypothetical mechanisms are illustrated in Fig. 9 to account for the enhanced and impeded In/Ga interdiffusion observed, respectively, in the present study and in polycrystalline CIGS.

First, the high Na concentration at the grain boundary may impede In/Ga interdiffusion due to the very reasons invoked in the literature, i.e., decreased concentration of $V_{Cu}$ defects[28]. Therefore, this widely accepted mechanism could still hold, but just at CIGS grain boundaries.

Second, the In or Ga enrichment (cf. refs. [50] and [57], respectively), on the other hand, could suggest a different solubility of In and Ga supercomplexes in CIGS that may depend on CGI ratio. If either of the two supercomplexes segregates preferentially at the grain boundaries, it could hinder In/Ga interdiffusion across the boundary due to Zener pinning.

This research shows that Na enhances In/Ga interdiffusion in CIGS films free from (high-angle) grain boundaries. The phenomenon is proposed to occur due to the formation of $\{[III_{Cu}V_{Cu}]^+[Na_{Cu}V_{Cu}]^-\}$ defect supercomplexes stabilised enthalpically by electrostatic attraction. The presence of such complex clusters is not proven but is consistent with SIMS and APT analyses.

These results add new experimental data to a crowded literature on the effects of Na doping in CIGS. Given the relevance of Ga depth profiling on the optoelectronic properties of CIGS solar cells, this study provides important physical insights into the interdiffusion of In and Ga and the role of Na. A clear

understanding of the role of Na on In/Ga interdiffusion will be beneficial for the development of more effective strategies for band-gap engineering and defect passivation. Therefore, the authors encourage computational testing of the hypotheses proposed.

## Methods

**Epitaxial film growth.** The $CuInSe_2$ films were grown epitaxially on 500 μm thick (100) GaAs wafers by metal-organic chemical vapour deposition (MOCVD). Cyclopentadienyl-copper-triethyl phosphine (CpCuTEP), trimethyl-indium (TMI) and ditertiarybutyl selenide (DtBSe) were decomposed at a substrate temperature of 470 °C, until a nominal thickness of 1000 nm was achieved, as monitored with an in-situ reflectometer. The film thickness varies macroscopically from the centre to the edge of the wafer, due to the deposition geometry. A Cu-poor CIS composition (Cu:In equals 0.87, as assessed by EDS) was obtained by adjusting the gas partial pressures to the following ratios p(DtBSe)/[p(CpCuTEP) + p(TMI)] = 26 and p(CpCuTEP)/p(TMI) = 0.7, keeping the CpCuTEP partial pressure to 0.45 mbar. A balancing flow of hydrogen for each source ensures a constant gas flow and a reactor pressure of 50 mbar throughout the deposition. More details on the epitaxial growth can be found elsewhere[58,59].

**Gas-phase post-deposition treatments.** The film characterisations were carried out on the as-deposited reference sample (untreated), and on samples having incurred thermal treatments at 570 °C for 30 min in the presence of solid sources of either elemental Se (Se-only) (100 mg) or mixtures of Se and $Na_2Se$ (Se+$Na_2Se$) (100 mg + 10 mg) (Alfa Aesar), as described in Supplementary Fig. 11. For all treatments a new as-deposited sample was used. A more detailed description of this kind of gas-phase PDT is provided elsewhere[25,59]. The process may generate small quantities of $H_2Se$ and should, therefore, be performed under fume hood to minimise safety risks. Attention is drawn to the risk of Na contamination during these experiments. Before the PDTs, the furnace assembly was subject to thorough cleaning by rinsing the quartz tubes and the graphite parts with running 18.2 MΩ deionized water and push-drying the excess water with a nitrogen flux. Due to the ubiquitous presence of Na, also gloves, tools and accessories were similarly rinsed. Lastly, the empty assembly was held at 800 °C for 48 h under constant vacuum ($<10^{-2}$ mbar) prior to the first PDT.

**Chemical, structural and optical characterisation.** Chemical and structural analyses performed in the films include: SIMS, scanning electron microscopy (SEM) with energy-dispersive X-ray spectroscopy (EDS), cross sectional nano-AES, electron back scatter diffraction (EBSD), X-ray diffraction (XRD) and X-ray RSM analyses.

SIMS measurements were performed with a Cameca SC-Ultra instrument using 1 keV $Cs^+$ ion bombardment, and all isotopic signals were normalised against the $Cs^+$ signal in order to compensate for any fluctuation of $Cs^+$ flux during the measurements, thus ensuring comparability of the results among the samples. The sputtering time was converted into depth by SEM cross sectional analysis. Multiple measurements were performed for statistical purposes. The Ga mole fraction in the films is estimated as the Ga/(In + Ga) SIMS signal ratio that serves as a reliable figure for qualitative comparison between the films. Although this does not provide a precise analytical quantification of the Ga content, the Ga/(In + Ga) SIMS signal ratio of a CIGS film with intentional 6% Ga has been measured and provides a reasonable value between 4 and 7%.

Nano-AES analysis (AES) was also performed to confirm the trends observed by top destructive SIMS. Taking advantage of the high lateral resolution of the AES technique, the in-depth composition was directly accessed by scanning along the sample section. Nano-AES analyses were performed deliberately on as-cleaved unpolished cross sections of the films, in order to prevent possible uncontrolled In/Ga interdiffusion during sample preparation by traditional manual or ion beam polishing. The surface was cleaned by a short and light $Ar^+$ sputtering sequence prior to analysis to reduce the surface carbon contamination that decreases the composition measurement accuracy (4 nm Auger electron escape depth). The probe tracking tool was employed to avoid drift positioning during acquisition. In order to minimise the effect of the as-cleaved topography, point by point analysis was performed at 15 kV, 10 nA, leading to independent composition determination along the cross section with a 15 nm spot size. Composition was determined using the sensitivity factors available in the constructor library yielding a composition with 1 to 5% accuracy.

The XRD measurements were carried out with PANalytical's MDP X'Pert-Pro instrument using a 1D PIXcel detector in the Bragg-Brentano configuration with a step size of 0.013° and 400 s acquisition time per step. A Cu counter-cathode operating at 45 kV and 40 mA was used, providing a 7 mm wide unfiltered X-ray beam on the sample resulting in a constantly irradiated/observed area of 14 mm². X-ray RSM studies were performed on all samples in order to verify the epitaxy of the films. RSMs were performed on a Bruker Discover D8 diffractometer set in a double axis geometry. In this set-up, the primary optics consists of a Göbel mirror, an automated absorber as well as a divergence slit of width 0.6 mm, whereas the secondary optics consisted of a 0.1 mm wide receiving slit as well as a 0D

scintillation detector. As no monochromator was used, both the Cu $K\alpha_1$ and $K\alpha_2$ wavelengths irradiating the sample contribute to the recorded RSMs. For all measurements, the goniometer radius was kept at 320 mm. The RSMs were recorded around the {511} reflection of the GaAs substrate.

Unlike untreated samples, Se-only and Se+$Na_2Se$ films were accessible by lift-off from the GaAs substrate, after embedding in epoxy resin, probably thanks to the strains and Kirkendall voids[27] formed at the CI(G)S/GaAs interfaces. Raman spectroscopy analysis from the back side of Se-only and Se+$Na_2Se$ epitaxial films were performed using a Renishaw spectrometer using a 532 nm laser excitation wavelength. Room temperature PL spectra were recorded on an InGaAs-array detector with a custom-made set-up using a 640 nm excitation laser wavelength with powers ranging from 1 to 100 mW and a spot diameter of 80 μm. PL measurements at 10 K were performed under the same conditions except the interposition of optical neutral density filters to reduce the incoming excitation intensity by four orders of magnitude.

The absorption coefficient of Se-only and Se+$Na_2Se$ films was derived from spectrophotometry, as the GaAs substrate is transparent in the wavelength region of interest. Transmittance and reflectance were measured in the near infrared range with a Perkin-Elmer Lambda 950 spectrophotometer equipped with an integrating sphere and a beam condenser allowing the measurements to be performed close to the areas analysed by SIMS. The absorption coefficient ($\alpha$) was calculated assuming a free standing and non-coherent film, so no effects of substrate and interference fringes were taken into account: $T = (1-r)^2 \times /(1-r^2X^2)$ and $R = r + TrX$, with $r$ being the air/film interface reflectivity and $X = \exp(-1/(\alpha d))$ with $d$ being the film thickness[60].

APT specimen and TEM lamella preparation was carried out using a dual-beam focused-ion-beam (FIB) (FEI Helios Nanolab 600i) following the lift-out technique described in ref. [61], on samples coated with 1 μm of Ni deposited by electron-beam evaporation. The APT tip shaping and TEM lamella thinning was performed using an acceleration voltage of 16 kV and currents of 150 and 50 pA for the Ga beam. To minimise beam damage due to Ga implantation a low acceleration voltage of 2–5 kV at a current of 5–7 pA for the Ga beam, i.e., the so-called low-kV FIB milling, was applied for final tip shaping and lamella thinning. APT analyses were performed using a local electrode atom probe (LEAP™ 5000XS, Cameca Instruments) operated at a base temperature of 60 K in laser pulsing mode, with a wavelength of 355 nm, ca. 10 ps pulse duration, and an energy of 5 pJ at a repetition rate of 250 kHz. The DC voltage was increased so as to maintain a detection rate of 10 ions per 1000 pulses. STEM-EDS analyses were carried out using a JEOL JEM-2200FS TEM equipped with a JEOL SDD EDX Detector 30, using an acceleration voltage of 200 kV; and with an FEI Titan Themis 60–300 X-FEG S/TEM equipped with a quad-silicon drift detector for EDS, using an acceleration voltage of 300 kV.

**Data availability.** The data which supports the findings of this work is available upon request from the corresponding author.

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

## Acknowledgements

The research leading to these findings has received funding from the Fonds National de la Recherche Luxembourg (FNR) through the GALDOCHS project (Gas-phase alkali doping of chalcogenide semiconductors, C14/MS/8302176) and from the German Research Foundation (DFG) (Contract GA 2450/1-1). F.W. and Su.S. acknowledge funding through the ODD (Optical detection of deep defects in chalcopyrite semiconductors) project. I.C.I. and I.P.A. acknowledge the FNR funding INTER/MOBILITY grant No. 15/9887562 and No. FNR-Inter2015/LRSF, respectively. We are grateful to Uwe Tezins, Andreas Sturm and Volker Kree for their support to the APT, TEM, SEM and FIB facilities at Max-Planck-Institut für Eisenforschung GmbH, and to Anaïs

Chauvière (LIST) for help with sample preparation. The initial findings of this study were presented at EMPA and Sapienza University of Rome in June 2016 as well as at the SPIE Optics + Photonic Conference held in San Diego in August 2016, D.C. wishes to thank the chairs and attendees for comments and feedback. D.C. wishes to thank the FNR for the CORE Junior track, as well Prof. A. Tiwari (EMPA) for his mentoring support.

## Author contributions

D.C. designed and carried out the PDT experiments, coordinated the research and wrote the manuscript. F.W. deposited the epitaxial films, T.S. performed the APT and STEM measurements, I.C.I. and Y.F. performed the XRD analyses, N.V. provided SIMS interpretation, C.S. performed the PL characterisation, E.V. carried out sample preparation and performed EBSD analysis, G.R. measured the transmission spectra, M.G. performed the Raman measurements, M.B. performed the nano-AES analysis, A.G.M. performed the STEM-EDS measurements of Se-only film. M.M. carried out the SEM-EDS analysis and B.E.A. performed the SIMS measurements. I.P.A., B.G., D.R., P.J.D. and Su.S. provided scientific support. All co-authors contributed with data interpretation.

## Additional information

**Competing interests:** The authors declare no competing financial interests.

