## [Peer Review File · Nature Communications]

Reviewers' comments:

Reviewer #1 (Remarks to the Author):

The authors claim that, contrary to previous reports on polycrystalline chalcopyrite films, Na enhances intragranular and hinders intergranular Ga/In diffusion in CuInSe₂. The results are novel, interesting, convincing, valuable, and they should certainly influence the field. I recommend accepting the paper after these questions are addressed:

1. Why did the films have 50% different thicknesses? Were the films different in any way other than thickness?
2. On lines 309 and 320 the authors discuss planar defects and grain boundaries in the samples, yet on line 350 it is claimed that the samples are free from grain boundaries. Please resolve the apparent contradictions.
3. Why were there no baseline Cu SIMS data? Was it normal to have such a large gradient in Cu composition throughout the thickness of your epitaxial CuInSe₂, or was that related to the high temperature Se PDT, or was it related to the Na PDT?
4. It is suggested on line 323 that the defect clusters may be unrelated to Na. Would comparing Figure 6 with baseline APT data resolve this uncertainty? If so, is there a reason the baseline data were not collected or published? If they were not collected, can the authors suggest future experimental work that would confirm or debunk the hypothesis of Na-enhanced intragranular Ga/In diffusion in CuInSe₂?

Regards,
Chris Muzzillo
July 18, 2017

Reviewer #2 (Remarks to the Author):

This manuscript is dealing with very fundamental issue of Na effects on In-Ga interdiffusion in CIGS thin film solar cells. I agree with the high values of this paper regarding novelty and insight in the fundamental aspect. However, I am doubt whether this is suitable for this journal since the scientific findings were not been proved by real solar cell devices. I would like to suggest that this manuscript is better to go more material oriented journals. Several comments which may help to improve the manuscript are suggested below.

1. Selenization with Na₂Se seems to play an important role in this study. TGA data of Se+Na₂Se needs to be presented.
2. Amount of Se and Se+Na₂Se used in the experiment should be noted.
3. Selenization was performed in H₂/N₂ environment. Are there any H₂Se gas generation problem and safety issues?
4. For practical aspects CIGS thin film should be prepared on a Mo-coated substrate in addition to a GaAs substrate with the same procedure. Please address about this issue.
5. In Figure 2(b) GGI should be presented by atomic %.
6. In depth profiling data is also required in Figure 2.

Reviewer #3 (Remarks to the Author):

The manuscript entitled " Na enhances In-Ga interdiffusion in Cu(In,Ga)Se₂ photovoltaic absorber "

reports the role of Na in enhancing the In/Ga inter-diffusion in epitaxial grown CIS films. In overall, this manuscript is well written and the experimental results are well described and well discussed logically. The author has done very thorough investigation using the various technologies such as secondary ion mass spectrometry (SIMS), nano-Auger electron spectroscopy (AES), atom probe tomograph, Raman etc. to conclude their observation of inter diffusion of In/Ga using extrinsic alkali (Na) doping of the film. These results definitely provide important insights about the role of alkali doping on these technologically important materials but the knowledge might be more restricted or beneficial to only epitaxial/monocrystalline CIGS absorber films which are free from grain boundaries. Most of the current state of art solar panel based on these compound semiconductors are polycrystalline in nature. Hence, the author can provide some explanation (with new experiments/data) on the relevance of these studies on the traditional polycrystalline absorber films. Does author have any physical insight about promoting the intergrain diffusion of In/Ga that would be beneficial for the conventional polycrystalline CIGS thin films? Also, does this inter-diffusion of In/Ga will be similar for other dopants such as K? Has author tried different dopant to see the rate of this inter-diffusion? Further, the author has done in-depth work on calculating the various strain in the system and correlate well with experimental data but showing SEM picture (in figure 3c) to confirm the strain effect on promoting the cracks is somewhat un-conclusive (as this can form during sample preparation). The author should add few additional low magnified SEM images of all three samples in SI to confirm this observation. Overall, this is an important piece of work and will be of interest to the readers of Nature Communication.

Reviewer #4 (Remarks to the Author):

Dear authors,

thank you for a highly interesting paper. In your study you have grown epitaxial CIGSe on GaAs substrates and annealed them in Se or Na+Se. You find that Ga diffuses into the CIGSe thin film to a higher extend if Na is present in the annealing environment. As a consequence you claim that - the widely accepted fact - a retarding effect on the interdiffusion of In/Ga in CIGSe thin films is not valid in all cases, or only under restrictions.

Your experimental work is extremely detailed and thorough and I studied it with much interest. However I find it hard to agree with the line of your argumentation, even though you make some very good points.

Together with this review you should receive the scan of a hand marked copy of the review. I understand that my handwriting is hard to read, so you will find numbers in circles along my markings and below I try to help by typing my remarks out. Please refer to my questions/remarks in the scanned document for the detailed review.

On a more general note I would like to stress the following: An annealing experiment is highly different from a growth experiment, i.e. annealing a CIGSe thin film in an atmosphere will rely on very different diffusion processes than those diffusion processes that happen during thin film growth. The widely accepted fact, that Na hinders In/Ga interdiffusion refers - in the majority of cases - to the case, when Na is present during growth. In this case it is also - to a certain extend - the CuInSe₂ and the CuGaSe₂ (or OVCs thereof) that are interdiffusing. In your case it is CIGSe and GaAs. The paper by H. Rodriguez-Alvarez I mention under remark 24 may help to understand what I mean. I am sure that your work has implications that are useful to be understood also for growth experiments, however I ask you to clarify and emphasize the difference and then use this as a starting point for your argumentation.

Then with respect to the style of the paper in my opinion you pick up some arguments and don't finish them. A lot of the discussion of your results is done in the "Results" and I would like to ask you to be more strict with the separation of the two or make clear where you start to interpret your measurements in the text. As a consequence the "Discussion" section is rather void and does not pick up on the topics you mention in the abstract. I would like to read in the discussion how you come to the conclusion that Na reduced the intergrain mobility of Ga, but not of Cu as it seems, why the Cu

concentration is highest at the front surface of the film, what are the implications of the stress in the films? What are the driving mechanisms for the diffusion happening in your opinion? You use a complex system of defect complexes for argumentation in the discussion. I think it would be good to help the reader by drawing a sketch of the diffusion process you suggest. Otherwise it is very hard to follow, the more so as it is new and still a matter of speculation.

And now to the detailed remarks:

- 1) I think your title is somewhat misleading and suggest to choose a more precise title e.g. "Na as cause for enhanced In-Ga-interdiffusion at epitaxial CIGSe/GaAs interfaces"
- 2) Where? "... at the GaAs/CIGSe interface."
- 3) Please explain Fig1. Is this result of the paper? Or a working hypothesis? I see a drawing on In-Ga-interdiffusion during CIGSe growth, but your paper is on Ga-outdiffusion from the GaAs substrate and its migration and effect in the CIGSe thin film.
- 4) What is the cause for this (different thickness)? Where the films of the same thickness before treatment? If yes why now thicker? Also, GGI or Ga-content (other side of page) ... please also give numbers in Fig (c) in the graph for ease of reading.
- 5) Here you claim higher accuracy for APT, however for Na in (a) SIMS and APT coincide ... hence I ask you to give a general account of the estimated error, i.e. a quantified discussion (I saw this is done to a certain extend in the method section).
- 6) Are these visible somewhere? How do you know they are there?
- 7) Possible but speculative. (This is certainly something for the discussion section in my opinion.)
- 8/9) Na induces considerable strain in chalcopyrites, especiall when added by PDT. Hence I am not confident that stress relaxation is an explanation for cracks in thin film after Na+Se treatment. Lit - H.F. Myers et al.; Journal of Crystal Growth 387 (2014) 36-40 and V. Lyahoritskaya et al.; Journal of Applied Physics 9177 (2002) 4205-4212.
- 10) Please indicate if tensile or compressive.
- 11) Please indicate if the "t" stands for.
- 12) Are the axes correctly labeled? I believe they may be exchanged. Please also give the indication for the position of CGSe.
- 13) Not really clear! I need to guess a lot.
- 14) please explain to the reader what you refer to by "fully strained". I don't understand.
- 15) I can't follow and don't understand the definition. Please add a sketch for ease of understanding.
- 16) Also the indiffusion of Na may add strain. As shown in Fig 2(a) Na diffuses into the bulk, and as you do claim that your film is epitaxial there are no grain boundaries, right? (I found this even more puzzling when I saw the APT. And then I reread your abstract and found the words "intergrain and intragrain", which alludes to grains and hence also grain boundaries being present. So I am asking you to discuss this topic in more detail.)
- 17) the bond length of Na-Se is a lot larger than Cu-Se ... -> so there are 2 processes Na-indiffusion->compressive strain and Ga-indiffusion->release of compressive strain (maybe?)
- 18) !? Why not 4.98? And no, according to the figure this is not the same for the CIS(1011) peak, tha is indicated.
- 19) Here I was confused, as you did not refer to values from one of the figures before, but to XRF measurements on the back of the samples, that where separately made. But maybe you can just refer to the SIMS profile? In line 248 and 253 - in my opinion you contradict yourself, as you claim SIMS cannot be used for compositional consideration and then you calculate NCN ... please justify.
- 20) ... and corresponding position ...
- 20a) (refers also to 16) You need to discuss, how this can be an epitaxial sample.
- 21) except for the peak at the interface as shown in Fig 5(c).
- 22) Is this on of the regions in Fig5? (This question is already answered later in the text.)
- 23) Se is offered in conjction with Na₂Se. Could this be related? How do you rule this out?
- 24) I could imagine that Indium diffuses out to the surface, drawn there by Se-overpressure and the presence of Na enhances this process. So it forms Cu-poor CIGSe. Ga follows to compensate for the In-loss in particular at the back. Cu then follows - somehow stress related - the Indium. Na goes to Cu-poor areas. Please compare: Rodriguez-Alvarez et al., Journal of Applied Physics 115 (2014) 204913. To counterargue you could claim a constant film thickness before and after Na₂Se anneal. But is this the case?
- 25) This really is part of the discussion.
- 26) Very different process, as Na is not present during growth here, but "only" during an anneal as PDT.

27) Film thickness before and after PDT known?

28) What does this depend on?

29) I think there is a huge difference between a PDT CIGSe sample and a sample that was exposed to Na supply during growth! The fact that everything seen in this work may be applicable only to PDT samples should be clearly stated somewhere prominent!

I do recommend publication after major revision! The experimental results and characterization efforts, also the learning from them are certainly worth the effort.

Best regards, Chris K.

Reviewer #1

The authors claim that, contrary to previous reports on polycrystalline chalcopyrite films, Na enhances intragranular and hinders intergranular Ga/In diffusion in CuInSe₂. The results are novel, interesting, convincing, valuable, and they should certainly influence the field. I recommend accepting the paper after these questions are addressed:

1. Why did the films have 50% different thicknesses? Were the films different in any way other than thickness?

The deposition geometry of the MOVPE yields epitaxial films with a macroscopic variation of thickness from the centre to the edge. Below we report an exemplary film thickness distribution determined by cross sectional SEM analysis on two parts of the sample employed for this study.

The specimens employed for the PDT study were 0.8 and 1.2 micron thick, so the difference in thickness is due to the variability of thickness distribution in the wafer. The specimens were taken from nearby areas of the same wafer to ensure the most consistent conditions, albeit at the expense of slightly different thickness. We believe this to be a better strategy than taking specimens from the same locations of different runs, because the compositional homogeneity within the same run is superior than between different runs.

The methods section of the manuscript has been modified in light of this point.

2. On lines 309 and 320 the authors discuss planar defects and grain boundaries in the samples, yet on line 350 it is claimed that the samples are free from grain boundaries. Please resolve the apparent contradictions.

We would like to thank the reviewer for spotting this inconsistency. Here it is important to make a distinction between grain boundaries present in polycrystalline films and the defects observed in the

present study. In order to investigate better these differences, we have performed dedicated scanning transmission electron microscopy (STEM) analyses (see below). These analyses reveal that the sample treated with Na contains a dense network of dislocations which may arrange to features as observed by APT or to low angle grain boundaries and planar defects such as stacking faults that by no means resemble conventional high angle grain boundaries typical of polycrystalline films.

Similar defects are also identified in the untreated film:

and in the Se-only film:

Furthermore, additional atom probe tomography (APT) analyses have been performed on the untreated and Se-only samples, for comparison with the Na-treated film. They indicate that defects such as stacking faults and/or dislocations are already present in the film before treatment and may incur Na-decoration/extension during the treatment.

The manuscript has now been modified based on the reviewer's comment and on the new experimental data acquired.

3. Why were there no baseline Cu SIMS data? Was it normal to have such a large gradient in Cu composition throughout the thickness of your epitaxial CuInSe_2 , or was that related to the high temperature Se PDT, or was it related to the Na PDT?

Although APT analysis reveals that nanoscale fluctuations of Cu concentration in the *untreated* film do exist (APT below), the SIMS data indicate that such gradients are much more pronounced in the *Na-treated* film (see below). As we believe that the large Cu gradients are a result of the Ga in-diffusion into the CIGS film, we have only presented the Na-treated data in the original submission.

APT untreated film [two figures are redacted here]

Cu and Na SIMS data of untreated, Se-only and Na-treated films (data from original submission):

Furthermore, additional STEM analysis has confirmed the presence of Kirkendall voids, as postulated in the original submission. Comparison between the density of voids and the Cu SIMS data suggests that the lack of Cu at the centre of the Na-treated film may partially be the result of the high density of Kirkendall voids in that region.

This prompted us to re-evaluate the SIMS data of Se+Na₂Se sample. We have then realized that in the original submission there was a mistake in the Cs-normalization of the SIMS data, due to human error in the data processing. Please note below the increase of Na and Cu SIMS signal at the interface with GaAs in the profiles of the original manuscript (dotted profiles, region highlighted by the red rectangle). In the original manuscript we could not provide an explanation to account for this effect. As a matter of fact it turned out that this increase is not real but it is due to a mistake in the original data processing. We have

now addressed this inconsistency (solid curves in the graph below). The corrected data shows no real Cu accumulation at the interface, which is consistent with the STEM and Raman analysis. We have updated Fig. 5 b-c accordingly.

The manuscript has been modified to include the missing SIMS data, the corresponding APT data and to rectify the mistake in the original SIMS normalization.

4. It is suggested on line 323 that the defect clusters may be unrelated to Na. Would comparing Figure 6 with baseline APT data resolve this uncertainty? If so, is there a reason the baseline data were not collected or published? If they were not collected, can the authors suggest future experimental work that would confirm or debunk the hypothesis of Na-enhanced intragranular Ga/In diffusion in CuInSe₂?

We would like to thank the reviewer for raising this point. We have included the suggested comparison in the revised manuscript. Please refer to point 2 and 3 above.

Reviewer #2:

This manuscript is dealing with very fundamental issue of Na effects on In-Ga interdiffusion in CIGS thin film solar cells. I agree with the high values of this paper regarding novelty and insight in the fundamental aspect. However, I am doubt whether this is suitable for this journal since the scientific findings were not been proved by real solar cell devices. I would like to suggest that this manuscript is

better to go more material oriented journals. Several comments which may help to improve the manuscript are suggested below.

1. Selenization with Na₂Se seems to play an important role in this study. TGA data of Se+Na₂Se needs to be presented.

We thank the reviewer for raising this point. We take this as an opportunity to highlight the novelty of the experimental approach employed. Na₂Se appears to evaporate incongruently, i.e. it releases atomic Na in the gas-phase when heated, along with small fractions of Na-Se molecules, as determined by Knudsen effusion mass spectrometry. We have not been able to detect any weight loss for the powder before and after the treatment. The absolute quantity of Na released is very small, yet sufficient to cause Na incorporation well below 1 at. %. These findings have been published in Scientific Reports 7 43266 (2017), DOI: 10.1038/srep43266.

2. Amount of Se and Se+Na₂Se used in the experiment should be noted. Addressed.

3. Selenization was performed in H₂/N₂ environment. Are there any H₂Se gas generation problem and safety issues?

We apologise for the lack of these details in the methods section. As per the safety issues, there is a thermodynamic driving force for Se to react with H₂ and form H₂Se. However, we have not studied the kinetics of reaction, so that we cannot provide a quantification of the H₂Se produced. In any case, the process is performed under fumehood to minimize safety issues.

These details have now been incorporated in the manuscript.

4. For practical aspects CIGS thin film should be prepared on a Mo-coated substrate in addition to a GaAs substrate with the same procedure. Please address about this issue.

All the CIGS studies reporting the effect of Na on In-Ga diffusion in the literature are focused on polycrystalline material, of which the great majority is on Mo substrates. In all cases Na appears to impede In-Ga interdiffusion in polycrystalline CIGS. These studies include our own recent paper published in Scientific Reports vol. 7, 43266 (2017), where a Na doping procedure similar to the one employed in the present study has been used (see excerpt below).

(a) EQE measurements of cells 1, 5 and 10 of sample Flux-NaCl showing an increasing CIGSe surface band gap (inset) due to (b) increasing gallium concentration away from the sodium source, as revealed by the Ga/(In + Ga) SIMS signal ratio.

Therefore, we think that this point has been already largely addressed by the community.

Rather, the goal of our current study is to understand if the impeding effect of Na on the diffusion of In and Ga is related to grain boundaries or not. The intent is to distinguish between the phenomena occurring in the bulk and at the grain boundaries of CIGS. This can only be addressed by looking at single crystalline material, such as epitaxial films, where there are no grain boundaries.

5. In Figure 2(b) GGI should be presented by atomic %.

The GGI ratio has no units by definition. Fig. 2b shows the GGI ratio expressed as the ratio of SIMS signals corresponding to the ^{69}Ga and ^{115}In isotopes, which is the simplest possible data treatment.

6. In depth profiling data is also required in Figure 2.

Figure 2 is entirely devoted to-depth profiling, so we are unsure about this point raised by the reviewer.

Reviewer #3:

The manuscript entitled " Na enhances In-Ga interdiffusion in Cu(In,Ga)Se₂ photovoltaic absorber " reports the role of Na in enhancing the In/Ga inter-diffusion in epitaxial grown CIS films. In overall, this manuscript is well written and the experimental results are well described and well discussed logically. The author has done very thorough investigation using the various technologies such as secondary ion mass spectrometry (SIMS), nano-Auger electron spectroscopy (AES), atom probe tomograph, Raman etc. to conclude their observation of inter diffusion of In/Ga using extrinsic alkali (Na) doping of the film. These results definitely provide important insights about the role of alkali doping on these technologically important materials but the knowledge might be more restricted or beneficial to only epitaxial/monocrystalline CIGS absorber films which are free from grain boundaries. Most of the current state of art solar panel based on these compound semiconductors are polycrystalline in nature. Hence, the author can provide some explanation (with new experiments/data) on the relevance of these studies on the traditional polycrystalline absorber films.

As discussed in point 4 of Reviewer #2, what makes epitaxial films different from polycrystalline films is the absence of grain boundaries. Therefore, epitaxial films should reveal the diffusion behaviour typical

of the bulk of CIGS grains in polycrystalline films, ensuring that such a behaviour is not concealed by the (normally) dominating effect of grain boundary diffusion.

The primary goal of this research is *not* to immediately improve the CIGS optoelectronic properties, but to improve the CIGS knowledge base.

Na is known to impede In-Ga interdiffusion in CIGS. Our conclusion is that this is only true at grain boundaries. Therefore, our work will have enduring repercussions on the CIGS technology if further studies will be able to engineer new strategies for grain boundary passivation that do not impede intergrain diffusion. However, such a task lies *beyond the scope of our manuscript* and cannot be reasonably implemented experimentally at review stage.

1. Does author have any physical insight about promoting the intergrain diffusion of In/Ga that would be beneficial for the conventional polycrystalline CIGS thin films?

We thank the reviewer for raising this point. It is a very good research question that is definitely worth pursuing. We think that the first step in this direction would be to understand why Na promotes bulk (*intragrain*) In-Ga diffusion in CIGS. We are currently working on this with a dedicated manuscript. Despite this, a possible underlying mechanism is outlined here and better emphasized in the revised discussion.

2. Also, does this inter-diffusion of In/Ga will be similar for other dopants such as K? Has author tried different dopant to see the rate of this inter-diffusion?

This is also a good research question. The role of other dopants such as K and Rb was proven to be important technologically (see e.g. Jackson et al. *physica status solidi (RRL)* 10, 8 583-586 (2016)). We have already started exploring the effect of other alkali metals. However, this requires a considerable amount of effort and we are still analyzing the data. Therefore, we cannot disclose the results yet.

3. Further, the author has done in-depth work on calculating the various strain in the system and correlate well with experimental data but showing SEM picture (in figure 3c) to confirm the strain effect on promoting the cracks is somewhat un-conclusive (as this can form during sample preparation). The author should add few additional low magnified SEM images of all three samples in SI to confirm this observation.

It is reasonable to wonder if SEM pictures are representative. Therefore, we have now included low magnification SEM images to corroborate our argument (see below). Additionally we also report new cross sectional SEM showing pronounced Kirkendall voiding in the Na-treated film (see below). Kirkendall voiding is also revealed by the new STEM analysis, see comment to point 3 of Reviewer #1.

We attribute the presence of cracks in the Na-treated film to a combined effect of compressive strain relief resulting from the replacement of In by smaller Ga, as well as to Kirkendall voiding.

We hope that the reviewer will convenue with us that the cracks are a consequence of the PDT process, as sample preparation was the same for all specimens (simple gluing to SEM stub with conductive tape).

Overall, this is an important piece of work and will be of interest to the readers of Nature Communication.

Reviewer #4

Dear authors,

thank you for a highly interesting paper. In your study you have grown epitaxial CIGSe on GaAs substrates and annealed them in Se or Na+Se. You find that Ga diffuses into the CIGSe thin film to a higher extend if Na is present in the annealing environment. As a consequence you claim that - the widely accepted fact - a retarding effect on the interdiffusion of In/Ga in CIGSe thin films is not valid in all cases, or only under restrictions.

Your experimental work is extremely detailed and thorough and I studied it with much interest. However I find it hard to agree with the line of your argumentation, even though you make some very good points.

Together with this review you should receive the scan of a hand marked copy of the review. I understand that my handwriting is hard to read, so you will find numbers in circles along my markings and below I try to help by typing my remarks out. Please refer to my questions/remarks in the scanned document for the detailed review.

We thank the reviewer for the encouraging words and also for the exceptional effort spent to review our manuscript.

On a more general note I would like to stress the following: An annealing experiment is highly different from a growth experiment, i.e. annealing a CIGSe thin film in an atmosphere will rely on very different diffusion processes than those diffusion processes that happen during thin film growth. The widely accepted fact, that Na hinders In/Ga interdiffusion referres - in the majority of cases - to the case, when Na is present during growth. In this case it is also - to a certain extend - the CuInSe₂ and the CuGaSe₂ (or OVCs thereof) that are interdiffusing. In your case it is CIGSe and GaAs. The paper by H. Rodriguez-Alvarez I mention under remark 24 may help to understand what I mean. I am sure that your work has implications that are useful to be understood also for growth experiments, however I ask you to clarify and emphasize the difference and then use this as a starting point for your argumentation.

We do agree with the reviewer that there are differences between annealing and growth experiments. This objection is a fine argument and we wish to thank the reviewer for pointing out that the difference is not highlighted sufficiently in the manuscript. We have now edited the manuscript to clarify the difference (red text below).

However, both growth and PDT experiments are subject to the same fundamental laws of diffusion. This fact is strengthened by the high diffusivity of Na in CIGS. Sodium has been reported to hinder In-Ga interdiffusion not only when it is present during the first stages of growth, but also when it is introduced later (e.g. D. Rudmann et al.; Thin Solid Films 431 –432 (2003) 37–40). Therefore, the new fundamental knowledge on CIGS diffusion generated by our study should be valuable also for growth experiments.

We think that our line of argument is similar to that proposed by Rodriguez-Alvarez et al. (Solar Energy Materials & Solar Cells 116(2013) 102–109) for polycrystalline CIGS. In the following “quoted” excerpt they claim that the alteration of Ga diffusivity would impact both Ga accumulation (a kinetically-limited phenomenon) and In/Ga interdiffusion (a process under thermodynamic control). It is just that our conclusions differ because our films are free from grain boundaries.

“In general terms, lowering the diffusivity of Ga both accentuates its accumulation near the back contact during the CIGSe formation and hampers the interdiffusion process. Our results suggest that in Cu-poor and Cu-rich films the presence of Na diminishes the overall diffusivity of Ga.”

The diffusion processes occurring during CIGS growth and PDTs are undoubtedly different. Nevertheless, they are subject to the same laws of diffusion. These results shed light on the mechanism of In/Ga interdiffusion in CIGS and suggest that the grain boundaries play a decisive role by acting as a barrier for intergrain diffusion of In and Ga in conventional polycrystalline CIGS films.

We also would like to stress the use of the verb *suggest* in the sentence above, implying that additional combined theoretical-experimental work would be needed to fully endorse our hypothesis. But most importantly, we hope that the reviewer will find the enhanced In/Ga interdiffusion observed with Na sufficiently convincing, being assessed by six independent analytical techniques.

Then with respect to the style of the paper in my opinion you pick up some arguments and dont finish them. A lot of the discussion of your results is done in the "Results" and I would like to ask you to be more strict with the separation of the two or make clear where you start to interpret your measurements in the text. As a consequence the "Discussion" section is rather void and does not pick up on the topics you mention in the abstract.

As a matter of fact, Nature Communications does not support a separate *conclusion* section. We introduced this *discussion* section because we felt the need to wrap up the necessarily lengthy *results* section. However, we agree with the reviewer that the paper deserves a better discussion. Therefore, we have edited the manuscript to make a clear distinction between results/interpretation and discussion and moved an edited version of Figure 1 to the new discussion section.

I would like to read in the discussion how you come to the conclusion that Na reduced the intergrain mobility of Ga, but not of Cu as it seems, why the Cu concentration is highest at the front surface of the film, what are the implications of the stress in the films? What are the driving mechanisms for the diffusion happening in your opinion? You use a complex system of defect complexes for argumentation in the discussion. I think it would be good to help the reader by drawing a schetch of the diffusion process you suggest. Otherwise it is very hard to follow, the more so as it is new and still a matter of speculation.

We understand and share the reviewers' interests and curiosity on the consequences of our findings.

We think it is erroneous to conclude from our findings that Na decreases Cu mobility. We believe that the increased Cu inhomogeneity after Na addition is rather the result of ODC phase segregation: Na increases Ga indiffusion to the extent that ODC must form out of the Cl(G)S due to phase equilibria considerations. The new STEM data even suggests that the ODC phase may form pseudo-topotactically, as no real grain boundaries are identified between Cl(G)S and ODC, although the interface is incoherent.

Unfortunately, it is not possible to answer all these relevant research questions in the present manuscript. However, we have now added a mechanistic description of the phenomenon based on all experimental evidence and clarified our argument on defect complex in the new discussion section (Fig. 7-8 below).

Figure 7 | Mechanistic description of In/Ga interdiffusion. Bright field STEM image of a cross section of $Se+Na_2Se$ sample and corresponding Ga, In and Cu EDS elemental maps.

Figure 8 | Effect of Na on In/Ga interdiffusion. Schematic representation of the effect of Na on (a) enhanced *intragrain* In/Ga interdiffusion, as seen here in CIGS films free from grain boundaries, and (b) hindered *intergrain* In/Ga interdiffusion, commonly observed in polycrystalline CIGS films. Na: green, In: blue, Ga: red.

And now to the detailed remarks:

1) *I think your title is somewhat misleading and suggest to choose a more precise title e.g. "Na as cause for enhanced In-Ga-interdiffusion at epitaxial CIGSe/GaAs interfaces"*

We agree that the suggested title is more specific. We formulated a very similar title at the beginning of the study. However, we do not find it appealing to a broad readership and would prefer to stick to the original version.

2) *... engineeredopper and gallium gradients*

Probably the reviewer refers to the 3 stage process, but there is no net Cu gradient in the final absorber. In any case, we think this detail is irrelevant for the point we want to make in this study.

Where? "... at the GaAs/CIGSe interface."

We think it is already clear from the context, we omit it to limit the words of the abstract.

3) Please explain Fig1. Is this result of the paper? Or a working hypothesis? I see a drawing on In-Ga interdiffusion during CIGSe growth, but your paper is on Ga-outdiffusion from the GaAs substrate and its migration and effect in the CIGSe thin film.

Upon reviewer's suggestion, this figure has been modified and moved to the new discussion section.

4) What is the cause for this (different thickness)? Where the films of the same thickness before treatment? If yes why now thicker? Please refer to Reviewer #1, point 1.

Also, GGI or Ga-content (other side of page) ... please also give numbers in Fig (c) in the graph for ease of reading. Addressed

5) Here you claim higher accuracy for APT, however for Na in (a) SIMS and APT coincide ... hence I ask you to give a general account of the estimated error, i.e. a quantified discussion (I saw this is done to a certain extend in the method section).

For APT we have higher chemical accuracy for a smaller field of view and with higher spatial resolution (in 3D) compared to SIMS. The error in terms of chemistry for APT depends on several factors related to the field-evaporation of the atoms, i.e. on the bonding, existence of defects, secondary phases, clusters, specimen shape, etc. and the measurement parameters as temperature, laser pulse energy and frequency. This all can alter the measured composition. Therefore, errors can differ from one specimen to another and even within the same specimen. The microstructure under investigation is very complex, hence, the error will likely change from region to region.

6) Are these visible somewhere? How do you know they are there?

7) Possible but speculative. (This is certainly something for the discussion section in my opinion.)

New evidence provided, please refer to reviewer #1 point 3 and reviewer #3 point 3.

8/9) Na induces considerable strain in chalcopyrites, especiall when added by PDT. Hence I am not confident that stress relaxation is an explanation for cracks in thin film after Na+Se treatment. Lit - H.F. Myers et al.; Journal of Crystal Growth 387 (2014) 36-40 and V. Lyahoritskaya et al.; Journal of Applied Physics 9177 (2002) 4205-4212.

We agree with the reviewer that Na can induce strain in CIS. As a matter of fact the level of strain depends on the concentration of Na to which the CIS is subject to. The literature cited clearly reveals Na-induced strain, but it is rather due to a redox chemical reaction occurring between the CIS and Na. Due to its low electronegativity, Na can strip Se from the CIS and form Na-Se species along with reduced Cu. This is the case in the paper by Myers et al. at high Na concentration (above 3%). However, at low concentrations (such as in our case, around one order of magnitude less), Na is soluble in CIS. This has been shown also recently by Forest et al. Journal of Applied Physics **121**, 245102 (2017) in single crystal CIS, where no strains have been reported. Indeed our own films are free of phases such as Na-Se and elemental Cu.

Rather, we think that the strains are induced by the asymmetric diffusion of In and Ga, which is exacerbated upon Na addition, leading to the concomitant Kirkendall void formation (please refer to reviewer #1 point 3 and reviewer #3 point 3).

10) Please indicate if tensile or compressive. Addressed

11) Please indicate if the "t" stands for.

t stands for tetragonal, but it has now been removed to improve clarity

12) Are the axes correctly labeled? I believe they may be exchanged. Please also give the indication for the position of CGSe. Thank you for spotting this inconsistency. Addressed.

13) Not really clear! I need to guess a lot.

As a matter of fact, this is a short general statement. It is not intended specifically for CIS but for any epitaxial film. It is meant to introduce the discussion of the crystallographic analysis. We think it is necessary in order to improve clarity to a wider audience.

14) please explain to the reader what you refer to by "fully strained". I don't understand.

15) I can't follow and don't understand the definition. Please add a schetch for ease of understanding.

The general statement discussed in the previous point has the purpose of defining strains and misfits in epitaxial films. In any case, we have now added explicitly that the film is a -oriented, meaning that the a parameter is the out-of-plane parameter, i.e. a lies perpendicular to the GaAs wafer. We have also addressed a mistake in the original graph and added two schematics as insets in the new graph to describe the two possible orientations of the CIS film on GaAs. Note that the a and c parameters of the CIS unit cell in the two cases is different (the effect is intentionally exaggerated), but the area (volume in 3D) is the same: volume conservation.

Note that the original Figure 3 containing both microstructural and crystallographic analyses has now been divided into two: Fig. 2 devoted exclusively to microstructure and Fig. 3 dedicated to the crystallography, with additional new EBSD analysis of the top surface of the films that confirm the epitaxial nature of the top surface of the films even after the treatments.

We have also performed an additional XRD analysis on the back side of Se+Na₂Se, and have included the new results in Figure 3. This has allowed us to further support the hypothesis that the additional phase detected is located at the interface with GaAs, as we observe an increase of the corresponding reflection. As to its identification, Raman and PL analyses are consistent with a Ga-rich ODC at the back of the film. The XRD reflection at 32deg 2theta matches well with the (200) reflections of CuGa₃Se₅ and CGS.

The new Figure 3 is reported below.

16) Also the indiffusion of Na may add strain. As shown in Fig 2(a) Na diffuses into the bulk, and as you do claim that your film is epitaxial there are no grain boundaries, right? (I found this even more puzzling when I saw the APT. And then I reread your abstract and found the words "intergrain and intragrain", which alludes to grains and hence also grain boundaries being present. So I am asking you to discuss this topic in more detail.)

17) the bond length of Na-Se is a lot larger than Cu-Se ... -> so there are 2 processes Na-indiffusion->compressive strain and Ga-indiffusion->release of compressive strain (maybe?)

We agree with the reviewer that Na indiffusion could also cause a variation of the out-of-plane parameter. Therefore, the text has been modified to acknowledge this possibility.

We also refer to point 2 of reviewer #1 for discussion on grain boundaries.

18) !? Why not 4.98? And no, according to the figure this is not the same for the CIS(1011) peak, that is indicated.

We apologise for the confusion caused. The H and L axes were inverted by mistake. This has been fixed in the new figure.

19) Here I was confused, as you did not refer to values from one of the figures before, but to XRF measurements on the back of the samples, that were separately made. But maybe you can just refer to the SIMS profile? In line 248 and 253 - in my opinion you contradict yourself, as you claim SIMS cannot be used for compositional consideration - and then you calculate NCN ... please justify.

We think that the EDS analysis specifically made on the back side of the sample provides a good estimate of GGI and CGI in this case. We have modified the text to make it clearer. Please refer also to Casino simulations in Fig. S11.

As per the SIMS, we thank the reviewer for spotting this inconsistency. The Na/(Na+Cu) ratio does not refer to atomic ratio, but to SIMS signal ratio. The atomic ratio is calculated only for the APT dataset and compared with the SIMS signal ratio in the same figure to provide a quantitative basis.

20) ... and corresponding position ... Addressed

20a) (refers also to 16) *You need to discuss, how this can be an epitaxial sample.*

The microstructure of this sample is extremely complex, but there are no real grain boundaries, as discussed in point 1 of reviewer #1. Please also refer to point 3 of reviewer #1, where new APT analysis of the reference untreated sample already shows defects and some Cu inhomogeneities.

21) *except for the peak at the interface as shown in Fig 5(c).*

Unfortunately, due to the high density of Kirkendall voids, it has proved impossible to reach the interface with GaAs while preparing APT tips. Therefore, it is not possible to compare Na and Cu concentrations at the back of the film with this technique.

22) *Is this on of the regions in Fig5? (This question is already answered later in the text.)*

23) *Se is offered in conjunction with Na₂Se. Could this be related? How do you rule this out?*

This is a good research question. We would be willing to investigate it further, but unfortunately we cannot address this point at the current stage.

24) *I could imagine that Indium diffuses out to the surface, drawn there by Se-overpressure and the presence of Na enhances this process. So it forms Cu-poor ClSe. Ga follows to compensate for the In-loss in particular at the back. Cu then follows - somehow stress related - the Indium. Na goes to Cu-poor areas. Please compare: Rodriguez-Alvarez et al., Journal of Applied Physics 115 (2014) 204913. To counterargue you could claim a constant film thickness before and after Na₂Se anneal. But is this the case?*

We appreciate the reviewer's suggestions. We do have some thermodynamic objections, though. For example, Se overpressure can destabilize a ternary chalcogenide such as ClS only at pressures sufficient to oxidize Cu(I) to Cu(II) and form CuSe phases, or in conjunction with higher alkali metal concentration, leading to separate Ak-In-Se phases, none of which have been observed here.

Overall, we think that drawing a detailed mechanism is too speculative at the moment, it lies outside the scope of the manuscript and would imply to exceed largely the word limit of Nature Communications. However, we are pursuing additional experimental work and a dedicated manuscript will follow.

The cited reference has been added to the manuscript to emphasize the difference in Ga diffusion due to kinetic and thermodynamic controls.

25) *This really is part of the discussion.*

We agree with the reviewer, this part has been moved to the discussion.

26) Very different process, as Na is not present during growth here, but "only" during an anneal as PDT.

The reviewer's suggestion to specify the different nature of the two processes has been included.

27) Film thickness before and after PDT known? Please refer to point 1 of reviewer #1.

28) What does this depend on?

The Auger quantification is performed using derivative spectra (to get rid of the background subtraction step) and directly depends on the sensitivity factors enabling to convert the peak-to-peak intensity into quantitative data. In this work we employed the sensitivity factors of the JEOL Library, determined by the constructor mainly on elemental or binary reference compounds whose chemical and physical properties were certified by additional analyses techniques (DRX, XPS, EDS...).

29) I think there is a huge difference between a PDT CIGSe sample and a sample that was exposed to Na supply during growth! The fact that everything seen in this work may be applicable only to PDT samples should be clearly stated somewhere prominent!

We think that any CIGS growers will be interested in these findings. The *broader* implications of the findings do not concern only PDT samples, but also samples where Na is present during the growth.

I do recommend publication after major revision! The experimental results and characterization efforts, also the learning from them are certainly worth the effort.

REVIEWERS' COMMENTS:

Reviewer #1 (Remarks to the Author):

The authors satisfactorily addressed all of the concerns raised by myself (Reviewer #1) and Reviewer #4 during the first round of review, and I recommend publication of the manuscript.

Reviewer #2 (Remarks to the Author):

The manuscript is properly revised in response to the reviewers' comments. I would like to recommend its acceptance.

Reviewer #3 (Remarks to the Author):

I have reviewed the authors reply to my initial review and I am happy that all suggested corrections have been made and these changes along with the changes made at the other reviewers suggestions make for a much improved submission and is suitable for publication.